# HP3O: Hybrid-Policy Proximal Policy Optimization with Best Trajectory

## Abstract

Proximal policy optimization (PPO) is one of the most popular state-of-the-art on-policy algorithms that has become a standard baseline in modern reinforcement learning with applications in numerous fields. Though it delivers stable performance with theoretical policy improvement guarantees, high variance and high sample complexity still remain critical challenges in on-policy algorithms. To alleviate these issues, we propose Hybrid-Policy Proximal Policy Optimization (HP3O), which utilizes a trajectory replay buffer to make efficient use of trajectories generated by recent policies. Particularly, the buffer applies the "first in, first out" (FIFO) strategy so as to keep only the recent trajectories to attenuate the data distribution drift. A batch consisting of the trajectory with the best return and other randomly sampled ones from the buffer is used for updating the policy networks. The strategy helps the agent to improve its capability on top of the most recent best performance and in turn reduce variance empirically. We theoretically construct the policy improvement guarantees for the proposed algorithm. HP3O is validated and compared against several baseline algorithms using multiple continuous control environments. Our code is available here.

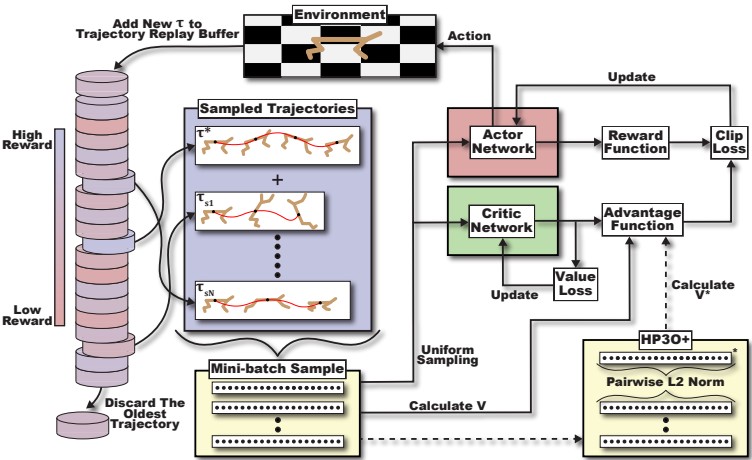

Figure 1: Schematic diagram of HP3O/HP3O+: (left side) the trajectory replay buffer takes a "first in, first out" (FIFO) strategy to keep only recent trajectories; batch consisting of the trajectory with the best return ($\tau^*$) and other randomly sampled ones from the buffer are used for updating the actor/critic networks (*off-policy* approach); (right side) model updating still follows the *on-policy* PPO method, hence, *hybrid-policy* PPO (HP3O); for HP3O+, $\tau^*$ is also used to update the advantage function

## 1 Introduction

Model-free reinforcement learning Liu et al. (2021) has demonstrated significant success in many different application areas, such as building energy systems Biemann et al. (2021), urban driving Toromanoff et al. (2020); Saxena et al. (2020), radio networks Kaur & Kumar (2020), robotics Polydoros

& Nalpantidis (2017), and medical image analysis Hu et al. (2023). In particular, on-policy reinforcement learning approaches such as proximal policy optimization (PPO) Schulman et al. (2017); Chang et al. (2023) provide stable performance along with theoretical policy improvement guarantees that involve a lower bound Kakade & Langford (2002) on the expected performance loss which can be approximated using the generated samples from the current policy. These guarantees are theoretically quite attractive and mathematically elegant, but the requirement of on-policy data and the high variance nature demands significant data to be collected between every update, inevitably causing the issue of high sample complexity and the behavior of slow learning.

Off-policy algorithms Zanette (2023); Prudencio et al. (2023), on the other hand, alleviate some of these issues as they can leverage a replay buffer to store samples that enable more efficient policy updates by reusing these samples. While the off-policy approach leads to better sample efficiency, it causes another problem called data distribution drift Zhang et al. (2020b); Lesort et al. (2021), and most studies Lillicrap et al. (2015); Dankwa & Zheng (2019) have just overlooked this issue. Furthermore, off-policy methods also suffer from high variance and even difficulty in convergence Lyu et al. (2020) due to the exploration in training. Mitigating this issue Bjorck et al. (2021) still remains challenging due to the high variations of stored samples in the traditional replay buffer design. However, it has been receiving considerable attention in recent studies Liu et al. (2020); Xu et al. (2019). Numerous previous attempts Zhang et al. (2021); Xu et al. (2020); Papini et al. (2018) took inspiration from supervised learning Wang et al. (2013); Johnson & Zhang (2013) and specifically made adjustments to the estimation of policy gradients to achieve variance reduction. However, this involves auxiliary variables and complex estimation techniques, resulting in a more complicated learning process. Another simple strategy to attenuate high variance is to leverage the advantage function involving a baseline Jin et al. (2023); Mei et al. (2022); Wu et al. (2018), which can be estimated by a parameterized model. Nevertheless, when the sampled data from the buffer has a large distribution drift, learning the parameterized model can be defective, triggering a poor advantage value. This naturally leads to the question:

*Can we design a hybrid-policy algorithm by assimilating the low sample complexity from off-policy algorithms into on-policy PPO for variance reduction?*

**Contributions.** We provide an affirmative answer to the above question. In this work, we blend off-policy and on-policy approaches to balance the trade-off between sample efficiency and training stability. Specifically, we focus primarily on mitigating underlying issues of PPO by using a trajectory replay buffer. In contrast with traditional buffers that keep appending all generated experiences, we use a "first in, first out" (FIFO) strategy to keep only the recent trajectories to attenuate the data distribution drift (as shown in Fig. 1). A batch consisting of the trajectory with the best return (a.k.a., best trajectory, $\tau^*$) and other randomly sampled ones from the buffer is used for updating the policy networks. This strategy helps the agent to improve its capability on top of the most recent 'best performance' and in turn to also reduce variance. Additionally, we define a new baseline which is estimated from the best trajectory selected from the replay buffer. Such a baseline evaluates how much better the return is by selecting the present action than the most recent best one, which intuitively encourages the agent to further improve the performance. More technical detail will be discussed in Section 4. Specifically, our contributions are as follows.

- We propose a novel variant of PPO, called Hybrid-Policy PPO (HP3O), that combines the advantageous features of on-policy and off-policy techniques to improve sample efficiency and reduce variance. We also introduce another variant termed HP3O+ that leverages a new baseline to enhance the model performance. Please see Table 1 for a qualitative comparison between the proposed and existing methods.

- We theoretically construct the policy improvement lower bounds for the proposed algorithms. HP3O provably shows a new lower bound where policies are not temporally correlated, while HP3O+ induces a value penalty term in the lower bound, which helps reduce the variance during training.

- We perform extensive experiments to show the effectiveness of HP3O/HP3O+ across a few continuous control environments. Empirical evidence demonstrates that our proposed algorithms are either comparable to or outperform on-policy baselines. Though off-policy techniques such as soft actor-critic (SAC) may still have better final returns for most tasks, our hybrid-policy algorithms have significantly more advantages in terms of run time complexity.

Table 1: Qualitative comparison with PPO and its relevant variants

| Method | T.B. | On/off-policy | T.G. |
|---|---|---|---|
| `PPO-Clip`Jin et al. (2023) | ✗ | ✗ | ✓ |
| `PTR-PPO`Liang et al. (2021) | ✓ | ✓ | ✗ |
| `GePPO`Queeney et al. (2021) | ✗ | ✓ | ✓ |
| `Policy-on-off PPO`Fakoor et al. (2020) | ✗ | ✓ | ✗ |
| `P3O`Chen et al. (2023) | ✗ | ✗ | ✗ |
| `Off-policy PPO`Meng et al. (2023) | ✗ | ✓ | ✓ |
| `HP3O(+) (ours)` | ✓ | ✓ | ✓ |

T.B.: trajectory buffer; T.G.: theoretical guarantee.

## 2 RELATED WORKS

**On-policy methods.** On-policy algorithms aim at improving the policy performance monotonically between every update. The work Kakade & Langford (2002) developing Conservative Policy Iteration (CPI) for the first time theoretically introduced a policy improvement lower bound that can be approximated by using samples from the present policy. In this regard, trust-region policy optimization (TRPO) Schulman et al. (2015) and PPO have become quite popular baseline algorithms. TRPO solves a trust-region optimization problem to approximately obtain the policy improvement by imposing a Kullback-Leibler (KL) divergence constraint, which requires solving a quadratic programming that may be compute-intensive. On the contrary, PPO achieves a similar objective by adopting a clipping mechanism to constrain the latest policy not to deviate far from the previous one during the update. Their satisfactory performance in different applications Hu et al. (2019); Lele et al. (2020); Zhang et al. (2022); Dutta & Upreti (2022); Bahrpeyma et al. (2023); Nguyen et al. (2024); Zhang et al. (2020a) triggers considerable interest in better understanding these methods Jin et al. (2023) and developing new policy optimization variants Huang et al. (2021). Albeit numerous attempts have been made in the above works, the high sample complexity due to the on-policy behavior of PPO and its variants still obstructs efficient applications to real-world continuous control environments, which demands the connection with off-policy methods.

**Off-policy methods.** To address the high sample complexity issue in on-policy methods, a common approach is to reuse the samples generated by prior policies, which was devised in Hester et al. (2018); Mnih et al. (2013). Favored off-policy methods such as deep deterministic policy gradient (DDPG) Lillicrap et al. (2015), twin delayed DDPG (TD3) Fujimoto et al. (2018) and soft actor-critic (SAC) Haarnoja et al. (2018) fulfilled this goal by employing a replay buffer to store historical data and sampling from it for computing the policy updates. As mentioned before, such approaches could cause data distribution drift due to the difference between the data distributions of current and prior policies. This work will include an implementation trick to address this issue to a certain extent. Kallus and Uehara developed a statistically efficient off-policy policy gradient (EOPPG) method Kallus & Uehara (2020) and showed that it achieves an asymptotic lower bound that existing off-policy policy gradient approaches failed to attain. Other works such as nonparametric Bellman equation Tosatto et al. (2020) and state distribution correction Kallus & Uehara (2020) were also done with off-policy policy gradient.

**Combination of on- and off-policy methods.** Making efficient use of on-policy and off-policy schemes is pivotal to designing better model-free reinforcement learning approaches. An early work merged them together to come up with the interpolated policy gradient Gu et al. (2017) for improving sample efficiency. Another work Fakoor et al. (2020) developed Policy-on-off PPO to interleave off-policy updates with on-policy updates, which controlled the distance between the behavior and target policies without introducing any additional hyperparameters. Specifically, they utilized a complex gradient estimate to account for on-policy and off-policy behaviors, which may result in larger computational complexity in low-sample scenarios. To compensate data inefficiency, Liang et al. Liang et al. (2021) incorporated prioritized experience replay into PPO by proposing a truncated importance weight method to overcome the high variance and designing a policy improvement loss function for PPO under off-policy conditions. A more recent work Chen et al. (2023) probed the insufficiency of PPO under an off-policy measure and explored in a much larger policy space to maximize the CPI objective. The most related work to ours is Queeney et al. (2021), where the

authors proposed a generalized PPO with off-policy data from prior policies and derived a generalized policy improvement lower bound. They utilized directly the past trajectories right before the present one instead of a replay buffer, which still maintains a weakly on-policy behavior. However, their method may suffer from poor performance in sparse reward environments.

## 3 PROBLEM FORMULATION AND PRELIMINARY

**Markov decision process.** In this context, we consider an infinite-horizon Markov Decision Process (MDP) with discounted reward defined by the tuple $\mathcal{M} = (\mathcal{S}, \mathcal{A}, p, r, \rho_0, \gamma)$, where $\mathcal{S}$ indicates the set of states, $\mathcal{A}$ signifies the set of actions, $p : \mathcal{S} \times \mathcal{A} \to \mathcal{S}$ is the transition probability function, $r : \mathcal{S} \times \mathcal{A} \to \mathbb{R}$ is the reward function, $\rho_0$ is the initial state distribution of environment, and $\gamma$ is the discount factor. In this study, the agent's policy is a stochastic mapping represented by $\pi : \mathcal{S} \to \mathcal{A}$. Reinforcement learning aims at choosing a policy that is able to maximize the expected discounted cumulative rewards $J(\pi) = \mathbb{E}_{\tau \sim \pi}[\sum_{t=0}^{\infty} \gamma^t r(s_t, a_t)]$, where $\tau \sim \pi$ indicates a trajectory sampled according to $s_0 \sim \rho_0$, $a_t \sim \pi(\cdot|s_t)$, and $s_{t+1} \sim p(\cdot|s_t, a_t)$. We denote by $d^\pi(s)$ a normalized discounted state visitation distribution such that $d^\pi(s) = (1-\gamma) \sum_{t=0}^{\infty} \gamma^t \mathbb{P}(s_t = s|\rho_0, \pi, p)$. Hence, the corresponding normalized discounted state-action visitation distribution can be expressed as $d^\pi(s, a) = d^\pi(s)\pi(s, a)$. Additionally, we define the state value function of the policy $\pi$ as $V^\pi(s) = \mathbb{E}_{\tau \sim \pi}[\sum_{t=0}^{\infty} \gamma^t r(s_t, a_t)|s_0 = s]$, the state-action value function, i.e., $Q$-function, as $Q^\pi(s, a) = \mathbb{E}_{\tau \sim \pi}[\sum_{t=0}^{\infty} \gamma^t r(s_t, a_t)|s_0 = s, a_0 = a]$, and the critical advantage function as $A^\pi(s, a) = Q^\pi(s, a) - V^\pi(s)$.

**Policy improvement guarantee.** The foundation of numerous on-policy policy optimization algorithms is built upon a classic policy improvement lower bound originally established in Kakade & Langford (2002). With different scenarios Schulman et al. (2015); Achiam et al. (2017); Dai & Gluzman (2021), the lower bound was refined to reflect diverse policy improvements, which can be estimated by using the samples generated from the latest policy. For completeness, we present in Lemma 1 the policy improvement lower bound from Achiam et al. (2017).

**Lemma 1.** *(Corollary 1 in Achiam et al. (2017)) Suppose that the current time step is $k$ and that the corresponding policy is $\pi_k$. For any future policy $\pi$, the following relationship holds true:*

$$J(\pi) - J(\pi_k) \geq \frac{1}{1-\gamma} \mathbb{E}_{(s,a) \sim d^{\pi_k}} \big[ \frac{\pi(a|s)}{\pi_k(a|s)} A^{\pi_k}(s, a) \big] - \frac{2\gamma C_{\pi_k}^\pi}{(1-\gamma)^2} \mathbb{E}_{(s,a) \sim d^{\pi_k}}[\delta(\pi, \pi_k)(s)], \quad (1)$$

*where $C_{\pi_k}^\pi = \max_{s \in \mathcal{S}} |\mathbb{E}_{a \sim \pi(\cdot|s)}[A^{\pi_k}(s, a)]|$ and $\delta(\pi, \pi_k)(s)$ is the total variation distance between the distributions $\pi(\cdot|s)$ and $\pi_k(\cdot|s)$.*

Lemma 1 implies that the policy improvement lower bound consists of the surrogate objective loss and the penalty term, which can be maximized by choosing a certain new policy $\pi_{k+1}$ to guarantee the policy improvement. However, directly maximizing such a lower bound could be computationally intractable if the next policy $\pi_{k+1}$ deviates far from the current one. Unless additional constraint is imposed such as a trust region in TRPO Schulman et al. (2015), which unfortunately requires a complex second-order method to solve the optimization problem. Hence, PPO developed a simple yet effective heuristic for achieving this.

**Proximal policy optimization.** PPO has become a default baseline in a variety of applications, as mentioned above. It is favored because of its strong performance and simple implementation with sound theoretical motivation given by the policy improvement lower bound. Intuitively, PPO attempts to constrain the new policy close to the present one with a *clipping* heuristic, which results in the most popular variant, PPO-clip Jin et al. (2023). Particularly, the following objective is solved at every policy update:

$$\mathcal{L}_k^{clip}(\pi) = \mathbb{E}_{(s,a) \sim d^{\pi_k}} \big[ \min(\frac{\pi(a|s)}{\pi_k(a|s)} A^{\pi_k}(s, a), \text{clip}(\frac{\pi(a|s)}{\pi_k(a|s)}, 1 - \epsilon, 1 + \epsilon) A^{\pi_k}(s, a)) \big], \quad (2)$$

where $\text{clip}(a, b, c) = \min(\max(a, b), c)$. The clipping function plays a critical role in this objective as it consistently enforces the probability ratio between the current and next policies in a reasonable range between $[1 - \epsilon, 1 + \epsilon]$. The outer minimization in Eq. 2 provides the lower bound guarantee for the surrogate loss in Eq. 1. In practice, one can set a small learning rate and a large number of time steps to generate sufficient samples to allow PPO to perform stably and approximate Eq. 2. However, due to its on-policy approach, high variance is a significant issue such that an extremely large number of samples may be required in some scenarios to make sure the empirical objective is able to precisely estimate the true objective in Eq. 2, which naturally causes the high sample complexity issue. This

motivates us to leverage off-policy techniques to alleviate such an issue, while keeping the theoretical policy improvement.

# 4 HYBRID-POLICY PPO (HP3O)

To achieve better sample efficiency of PPO, historical samples generated by previous policies are reused for policy updates, as done in off-policy algorithms. This inevitably results in a distribution drift between policies, which essentially disproves the policy improvement lower bound in Lemma 1. In this context, to fix this issue, we will extend Lemma 1 to assimilate off-policy samples in a principled manner to derive a new policy improvement lower bound that works for our proposed algorithm, HP3O. HP3O (and its variant HP3O+) takes a *hybrid* approach that effectively synthesizes on-policy trajectory-wise policy updates and off-policy trajectory replay buffers. Algorithm 1 shows

---

**Algorithm 1** HP3O(+)

---

1: **Input:** initializations of $\theta_0$, $\phi_0$, and trajectory replay buffer $R$, the number of episodes $K$, the number of time steps in each episode $T$, the number of epochs for updates $E$
2: **for** $k = 1, 2, ..., K$ **do**
3:      Run policy $\pi_{\theta_k}$ to generate a trajectory $\tau = (s_0, a_0, r_1, s_1, ..., s_{T-1}, a_{T-1}, r_T)$
4:      Append $\tau$ to $R$ and discard the oldest one $\tau^-$             ▷ FIFO strategy
5:      Sample a random minibatch $\mathcal{B}$ from the trajectory replay buffer $R$
6:      Select the best action trajectory $\tau_k^*$ from the trajectory replay buffer and add it to $\mathcal{B}$
7:      **for** each trajectory $j = 1, 2, ..., |\mathcal{B}|$: **do**
8:          **for** $t = 0, 1, ..., T-1$ **do**
9:              $G_t^j = \sum_{l=t+1}^{T} \gamma^{l-t-1} r_l^j$
10:          **end for**
11:      **end for**
12:      Compute advantage estimates $\hat{A}_t^{\pi_k} = G_t - V_\phi(s_t)$             ▷ HP3O
13:      Compute $V^{\tau_k^*}(s_t)$ using $\tau_k^*$ and advantage estimates $\hat{A}_t^{\pi_k} = G_t - V^{\tau_k^*}(s_t)$      ▷ HP3O+
14:      **for** each epoch $e = 1, 2, ..., E$ **do**
15:          Compute the clipping loss Eq. 2
16:          Compute the mean square loss $\mathcal{L}^V(\phi) = -\frac{1}{T} \sum_{t=0}^{T-1} (G_t - V_\phi(s_t))^2$
17:          Update $\pi_{\theta_k}$ with $\nabla_\theta \mathcal{L}^{clip}(\theta)$ by Adam
18:          Update $V_{\phi_k}$ with $\nabla_\phi \mathcal{L}^V(\phi)$ by Adam
19:      **end for**
20: **end for**
21: **return** $\pi_{\theta_K}$ and $V_{\phi_K}$

---

the algorithm framework for HP3O and HP3O+ (blue line represents the only difference for HP3O+). We denote the actor and critic by $\theta \in \mathbb{R}^m$ and $\phi \in \mathbb{R}^n$ respectively such that the parameterized policy function is $\pi_\theta$ and the parameterized value function is $V_\phi = \mathbb{E}_{\tau \sim \pi_\phi}[\sum_{l=t}^{T} \gamma^{l-t} r(s_l, a_l)|s_t]$. Denote by $\tau_k^* = \operatorname{argmax}_{\tau \in R} \sum_{t=0}^{T} \gamma^t r(s_t, a_t)$ the best action trajectory selected from the replay buffer $R$ at the current episode $k$.

In most existing off-policy algorithms, the size of the replay buffer is fixed with a large number to ensure that a diverse set of experiences is captured. With this approach, though the random minibatch sampling allows the agent to learn from past experience, a large-size replay buffer may cause significant data distribution drifts. Additionally, a large replay buffer means that it takes more time for the buffer to fill up, especially in environments requiring extensive exploration. Hence, we apply the FIFO strategy and discard old trajectories empirically to attenuate the issue (Line 4 in Algorithm 1). The recently proposed off-policy PPO Meng et al. (2023) indeed uses off-policy data, but it does not employ a trajectory buffer as we do. In our approach, the trajectory buffer is an essential component because it allows us to store and process complete sequences of state-action pairs (trajectories) rather than isolated transitions. This will preserve the temporal coherence and enhance stability. Line 5 is to sample from the trajectory replay buffer $R$, which is different from the reuse of $N$ samples generated from prior policies in Queeney et al. (2021), where the past immediate sample trajectories were used without random sampling. We note that a replay buffer in the proposed algorithm enhances the agent's performance by providing access to a more diverse

set of experiences and highlighting the most impactful trajectories. Line 6 signifies the core part of HP3O as the best action trajectory $\tau_k^*$ indicates the best return starting from state $s_t$ within the buffer. Line 7 through Line 12 calculate the rewards to go for each time step $t$ in each trajectory and obtain the total reward to go at each time step over all trajectories. One may wonder how to calculate the return $G_t$ if trajectories have varying lengths in some environments. In this work, we store different lengths of trajectories directly in the buffer and do not pad them. This approach preserves the natural variation in trajectory lengths that can occur in different environments. Although the length differ, we still compare the returns of these trajectories to identify the best one while ensuring the comparison remains consistent and fair. Particularly, line 13 is a key step in the proposed HP3O+. $V^{\tau_k^*}(s_t) = \mathbb{E}_{\tau_k^* \sim \pi_k}[\sum_{l=t}^{T} \gamma^{l-t} r(s_l, a_l)|s_t]$ induced by the current best action trajectory $\tau_k^*$ sets the best state value among all trajectories from $R$. $\hat{A}_t^{\pi_k}$ in Line 13 signifies how much better the return $G_t$ is by taking action $a_t$ than the best value we have obtained most recently. Intuitively, this "encourages" the agent to improve its performance in the next step on top of $V^{\tau_k^*}(s_t)$. While $V^{\tau_k^*}(s_t)$ can be theoretically calculated as above, in practice, to make sure that there always exists a best value for use, $V^{\tau_k^*}(s_t)$ is calculated by using a norm distance between the current trajectory and best trajectories to ensure $V^{\tau_k}$ has the best return since $s_t$. If the reward to go from $s_t$ in the best trajectory is lesser, the current trajectory is used to replace the best one for $V^{\tau_k^*}(s_t)$ calculation. Please see the Appendix for more details about the data structures of the proposed algorithms.

**Remark 1.** *We remark on the sampling method adopted in this work to obtain the trajectories apart from the best trajectory for update. We begin by randomly sampling a set of trajectories from our trajectory buffer. This set is specifically designed to include the best action trajectory, with the remaining trajectories selected randomly from the buffer. From the set of trajectories obtained by random sampling, we then apply uniform sampling. The resulting minibatch is used for training. This approach balances leveraging high-performing trajectories while maintaining exploration across the broader trajectory space, helping to reduce the risk of overfitting. However, we recognize that assigning a score to trajectories based on the loss function could offer additional benefits. Prioritizing trajectories Hou et al. (2017) that result in higher losses could help the agent focus on challenging experiences, potentially improving learning efficiency by addressing areas where the policy requires more refinement. This could also help in stabilizing training by emphasizing learning from mistakes, thereby potentially reducing the variance in policy updates. In fact, integrating a prioritized experience replay (PER) strategy could be a promising direction for future work.*

## 5 THEORETICAL ANALYSIS

This section presents a theoretical analysis of the proposed HP3O and HP3O+. We first derive a new policy improvement lower bound for HP3O and then present a different bound for HP3O+ to indicate the value penalty term. All proofs are deferred to the Appendix. To incorporate prior policies in the policy improvement lower bound, we need to extend the conclusion in Lemma 1, which quantifies the improvement for two consecutive policies. In Queeney et al. (2021), policies prior to the present policy $\pi_k$ in chronological order were used. However, in our study, this order has been broken due to the random sampling from the replay buffer, which motivates us to derive a relationship among the current, future, and prior policies independent of the chronological order. Before the main result, we first present an auxiliary technical lemma.

**Lemma 2.** *Consider a current policy $\pi_k$, and any reference policy $\pi_r$. For any future policy $\pi$,*

$$J(\pi) - J(\pi_k) \geq \frac{1}{1-\gamma}\mathbb{E}_{(s,a)\sim d^{\pi_r}}\Big[\frac{\pi(a|s)}{\pi_r(a|s)}A^{\pi_k}(s,a)\Big] - \frac{2\gamma C_{\pi_k}^{\pi}}{(1-\gamma)^2}\mathbb{E}_{s\sim d^{\pi_r}}[\delta(\pi,\pi_r)(s)], \quad (3)$$

*where $C_{\pi_k}^{\pi}$ and $\delta(\pi,\pi_r)(s)$ are defined as in Lemma 1.*

**Remark 2.** *Lemma 2 implies that now the visitation distribution, the probability ratio of the surrogate objective, and the maximum value of the total variation distance depend on the reference policy $\pi_r$, which essentially extends Lemma 1 to a more generalized case. However, the improvement is still for the two consecutive policies $\pi_k$ and $\pi$ as the advantage function in the surrogate objective and $C_{\pi_k}^{\pi}$ rely on the latest policy $\pi_k$. Lemma 2 does not necessarily require $\pi_r$ to be the last policy prior to $\pi_k$ as in Queeney et al. (2021), which paves the way for establishing the policy improvement for $|\mathcal{B}|$ prior policies sampled randomly from the replay buffer $R$.*

**Theorem 1.** *Consider prior policies $|\mathcal{B}|$ randomly sampled from the replay buffer $R$ with indices $i = 0, 1, ..., |\mathcal{B}| - 1$. For any distribution $v = [v_1, v_2, ..., v_{|\mathcal{B}|}]$ over the $|\mathcal{B}|$ prior policies, and any*

*future policy $\pi$ generated by HP3O in Algorithm 1, the following relationship holds true*

$$J(\pi) - J(\pi_k) \geq \frac{1}{1-\gamma}\mathbb{E}_{i\sim v}[\mathbb{E}_{(s,a)\sim d^{\pi_i}}[\frac{\pi(a|s)}{\pi_i(a|s)}A^{\pi_k}(s,a)]] - \frac{\gamma C^{\pi}_{\pi_k}\epsilon}{(1-\gamma)^2}, \tag{4}$$

*where $C^{\pi}_{\pi_k}$ is defined as in Lemma 1.*

**Remark 3.** *It is observed that the conclusion from Theorem 1 is similar to one of the main results in Queeney et al. (2021). The significant difference is that $\pi_i$ is not the same as $\pi_{k-i}$ in Queeney et al. (2021). It is technically attributed to Lemma 2, where the reference policy $\pi_r$ may not have a close temporal relationship with $\pi_k$. Also, the advantage function has not been changed yet. Empirically speaking, for each minibatch $\mathcal{B}$, we have added the best trajectory in it, which essentially expedites the learning process. Additionally, Theorem 1 has an extra expectation operator over multiple trajectories on the first term of the right side in Eq. 4, leading to the smaller variance, compared to only one trajectory in Lemma 1. We would also like to point out that Theorem 1 shows the policy improvement lower bound by sampling a mini-batch of trajectories associated with prior policies from the buffer, which is consistent with what has been done in Algorithm 1. In HP3O+, we use it as a baseline to replace $V_\phi(s)$ and have surprisingly found that this leads to an extra term that penalizes the state value to reduce the variance.*

We first define $\hat{A}^{\pi}(s,a) = Q^{\pi}(s,a) - V^{\pi^*}(s)$ and $G^{\pi}(s) = V^{\pi^*}(s) - V^{\pi}(s)$. It is immediately obtained that $A^{\pi}(s,a) = \hat{A}^{\pi}(s,a) + G^{\pi}(s)$. Hence, if we utilize the state value induced by the best trajectory at the moment as the baseline, there exists a *value gap* $G^{\pi}(s)$ between $A^{\pi}(s,a)$ and $\hat{A}^{\pi}(s,a)$. One may argue that the advantage $\hat{A}^{\pi}(s,a)$ is negative all the time, which implies the present action is not favorable such that the new policy should be changed to yield a lower probability for the current action and state. However, this is not always true as $V^{\pi^*}(s)$ is not the *globally* optimal value, while it is approximately the optimal value up to the current time step over the last $|\mathcal{B}|$ episodes. The motivation behind $\hat{A}^{\pi}(s,a)$ is that the new baseline $V^{\pi^*}(s)$ becomes the driving force to facilitate the performance improvement between every update. We are now ready to state the policy improvement lower bound with the new baseline as follows.

**Lemma 3.** *Consider a current policy $\pi_k$, and any reference policy $\pi_r$. For any future policy $\pi$,*

$$J(\pi) - J(\pi_k) \geq \frac{1}{1-\gamma}\mathbb{E}_{(s,a)\sim d^{\pi_r}}[\frac{\pi(a|s)}{\pi_r(a|s)}\hat{A}^{\pi_k}(s,a)] - \frac{2\gamma\hat{C}^{\pi}_{\pi_k}}{(1-\gamma)^2}\mathbb{E}_{s\sim d^{\pi_r}}[\delta(\pi,\pi_r)(s)]$$
$$- \frac{2\gamma C^{\pi_k}}{(1-\gamma)^2}\mathbb{E}_{s\sim d^{\pi_r}}[\delta(\pi,\pi_r)(s)], \tag{5}$$

*where $\hat{C}^{\pi}_{\pi_k} = \max_{s\in\mathcal{S}}|\mathbb{E}_{a\sim\pi(\cdot|s)}[\hat{A}^{\pi_k}(s,a)]|$, $\delta(\pi,\pi_r)(s)$ is defined as in Lemma 1, $C^{\pi_k} = \max_{s\in\mathcal{S}}|V^{\pi_k^*}(s) - V^{\pi_k}(s)|$.*

With Lemma 3 in hand, we have another main result in the following.

**Theorem 2.** *Consider prior policies $|\mathcal{B}|$ randomly sampled from the replay buffer $R$ with indices $i = 0, 1, ..., |\mathcal{B}| - 1$. For any distribution $v = [v_1, v_2, ..., v_{|\mathcal{B}|}]$ over the $|\mathcal{B}|$ prior policies, and any future policy $\pi$ generated by HP3O+ in Algorithm 1, the following relationship holds true*

$$J(\pi) - J(\pi_k) \geq \frac{1}{1-\gamma}\mathbb{E}_{i\sim v}[\mathbb{E}_{(s,a)\sim d^{\pi_i}}[\frac{\pi(a|s)}{\pi_i(a|s)}\hat{A}^{\pi_k}(s,a)]] - \frac{\gamma\hat{C}^{\pi}_{\pi_k}\epsilon}{(1-\gamma)^2} - \frac{\gamma C^{\pi_k}\epsilon}{(1-\gamma)^2}, \tag{6}$$

*where $\hat{C}^{\pi}_{\pi_k}$ and $C^{\pi_k}$ are defined as in Lemma 3.*

**Remark 4.** *Theorem 2 describes the policy improvement lower bound for HP3O+, which provides the theoretical guarantees when reusing trajectories generated by prior policies rigorously. The extra term on the right-hand side $\frac{\gamma C^{\pi_k}\epsilon}{(1-\gamma)^2}$ in the above inequality is not the penalty term between two policies, while it is a value gap between the current state value and the most recent best value. As $V^{\pi_k^*}(s)$ is time-varying, this acts as a "guide" to the current one $V^{\pi_k}$ not deviating too far away from $V^{\pi_k^*}(s)$. Equivalently, the term $\frac{\gamma C^{\pi_k}\epsilon}{(1-\gamma)^2}$ can be regarded as a regularization from the critic network, which assists in enhancing the overall agent performance and reducing the variance. We also include some technical discussion regarding whether our approach will cause overfitting and the adoption of the worst trajectories in Appendix A.2 and A.3.*

**Remark 5.** *The proposed HP3O algorithm and its variant have resorted to data randomly sampled from multiple policies in the training batch $\mathcal{B}$ that is prior to $\pi_k$ for the policy update. Thus, there exist multiple updates compared to the vanilla PPO, which only makes one policy update from $\pi_k$*

*to $\pi_{k+1}$. In this study, we aim to show how the off-policy sample reuse significantly affects the original sample efficiency PPO has. Though the direct sample complexity improvement analysis can be significantly beneficial to provide a solid theoretical foundation for the proposed algorithms, a thorough investigation of this aspect is out of the scope of this study. For instance, to arrive at an $\varepsilon$-optimality for policy gradient-based algorithms, a few works Zhong & Zhang (2024); Zanette et al. (2021); Dai et al. (2023); Sherman et al. (2023) have revealed the exact complexity with respect to $\varepsilon$, but only for MDPs with linear function approximation. The exact sample complexity analysis for the off-policy PPO algorithm with nonlinear function approximation still remains extremely challenging and requires a substantial amount of non-trivial efforts. Therefore, in this paper, we instead disclose the impact of off-policy sample reuse on the tradeoff between sample efficiency and learning stability. Please see Appendix A.4 for more details. Additionally, we also present a theoretical result in Appendix A.5 to reveal that HP3O+ increases updates in the total variational distance of the policy throughout training, given the same sample size, when it is compared to HP3O.*

## 6 EXPERIMENTS

The experimental evaluation aims to understand how the sample complexity and stability of our proposed algorithms compare with existing baseline on-policy and off-policy learning algorithms. Concretely, we conduct the comparison between our methods and prior approaches across challenging continuous control environments from the Gymnasium benchmark suite Brockman et al. (2016). While easy control tasks can be solved by various algorithms, the more complex tasks are typically sample intensive with on-policy algorithms Schulman et al. (2017). Additionally, the high variance of the algorithms negatively impacts stability and convergence. Furthermore, though some off-policy algorithms enjoy high sample efficiency, the actual run time can be impractically large, which impedes its applications to real-world tasks. As our proposed hybrid-policy learning algorithms are developed on top of PPO, we mainly compare our methods to PPO, another popular on-policy method A2C Peng et al. (2018), and three other relevant off-policy PPO approaches, including P3O Chen et al. (2023) (a modification of PPO to leverage both on- and off-policy principles), GEPPO Queeney et al. (2021), and Off-policy PPO (abbreviated as OffPolicy) Meng et al. (2023). We acknowledge that SAC, a fully off-policy algorithm, may achieve comparatively higher returns in most of the continuous control problems at the expense of much longer training time and with careful hyperparameter tuning. Hence, we also compare with SAC in terms of variance reduction and run time complexity. As shown in Table 1, there are other off-policy versions of PPO, such as Policy-on-off PPO Fakoor et al. (2020). However, the corresponding code base lacks a complete implementation to reproduce their results, which is evident in their code where the actor head for Mujoco environments is not implemented. Moreover, making the code functional for our purpose would require extensive effort, as it is built on MXNet, a deprecated open-source project. The above limitations have prevented us from performing head-to-head comparisons. More details about hyperparameter settings are deferred to the Appendix.

### 6.1 COMPARATIVE EVALUATION

Figure 2 shows the total average return during training for A2C, PPO, P3O, GEPPO, OffPolicy, HP3O, and HP3O+. Each experiment includes five different runs with various random seeds. The solid curves indicate the mean, while the shaded areas represent the standard deviation over the five runs. Clearly, the results show that, overall, both HP3O+ and HP3O are comparable to or outperform all baselines across diverse tasks with smaller variances, which supports our theoretical claims. For instance, in the HalfCheetah environment, our methods demonstrate a sharper average slope compared to the baseline, particularly in the later stages of training, where other baselines show a more flattened curve. This indicates that our method continues to learn effectively with fewer samples. In the Hopper environment, P3O performs slightly better than HP3O but at the cost of extremely large reward variance, indicating an unstable training process. However, HP3O+ significantly dominates in the latter phase with a much smaller variance. In the Swimmer environment, while A2C and P3O learn slowly and make almost no progress, HP3O and HP3O+ achieve the similarly highest reward with very low variance, as suggested by Remark 3. Notably, OffPolicy ranks second in terms of performance, but with the cost of extremely high variance. Additionally, OffPolicy shows notably strong performance in the Walker environment. This is primarily attributed to the adoption of a new clipped surrogate that iteratively resorts to off-policy data to progress during training. Generally, our proposed methods learn more stably than all baselines by dequeuing the buffer to suppress the instability caused by data distribution drift in most environments. Overall, HP3O+ excels HP3O in most environments, with also variance reduction particularly in the latter training phase. As the learning trajectories are always around the best trajectory from the buffer. Essentially, the empirical

evidence supports our theoretical results in Theorem 2 and Theorem 5, which show that HP3O+ enables larger updates in the total variational distance of the policy, given the same number of changes to the policy. Additional results are included in the Appendix, including Table 2 to showcase rewards at or close to the converged stage.

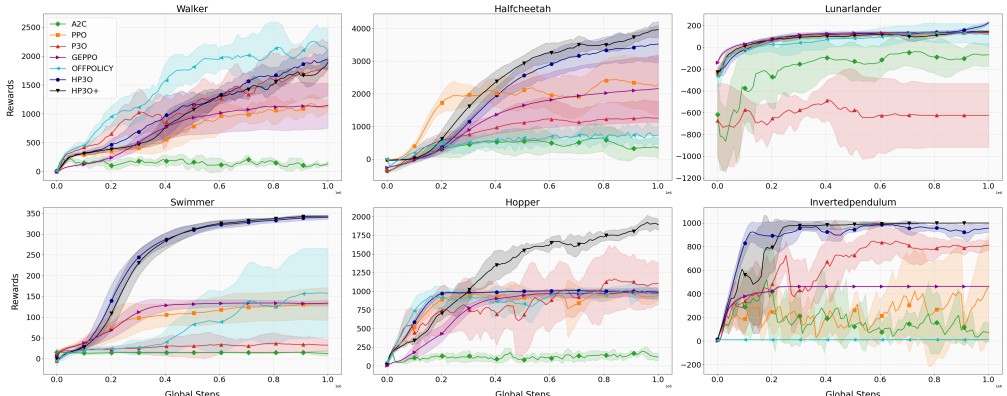

Figure 2: Training curves (over 1M steps) on continuous control benchmarks. HP3O+ (black) performs consistently across all tasks and is comparable to or outperforming other baseline methods.

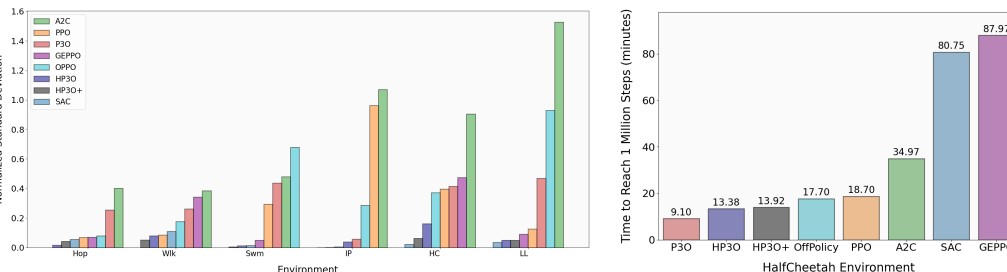

(a) Normalized Standard Deviation among different methods for various environments.

(b) Runtime for HalfCheetah Environment among different methods

Figure 3: Comparison of Normalized Standard Deviation and Runtime for 1 million steps.

## 6.2 ABLATION STUDY

The experimental results in the previous section imply that algorithms based on the hybrid-policy approach can outperform the conventional on-policy methods on challenging control tasks. In this section, we further compare all policy optimization algorithms to SAC for variance reduction and run time complexity. We also inspect the robustness of the algorithms against variations of trajectories.

**Variance.** Figure 3a shows the comparison of the relative standard deviation of the ultimate average return (at 1M steps) for different algorithms. It suggests that, on average, HP3O+ achieves the lowest relative standard deviation (which is the ratio of the standard deviation to the average reward over five runs at the last step). This implies that hybrid-policy algorithms have more advantages in regularizing the learning process to maintain stability compared to typical on-policy algorithms. Intuitively, as the policy and environment change over time, the use of replay buffers helps mitigate this issue by providing a more stationary training dataset. The buffer contains a mix of experiences collected under different policies, instead of the only current policy from PPO, which helps in reducing the variance in updates. SAC attains a relatively small standard deviation according to Figure 3a (also, on average, the maximum reward reported in the Appendix). This is not surprising since the maximum entropy principle can significantly help meaningful exploration to achieve the highest return. However, this comes at the cost of runtime complexity.

**Run time complexity.** As shown in Figure 3b, the run time for all algorithms is presented (all methods are implemented with the same hardware). Both GEPPO and SAC require much more run time to explore and then converge, which may impede its applications to solving real-world problems. P3O achieves the lowest run time complexity while performing worse than HP3O and

HP3O+. However, our proposed approaches take approximately the same training time as PPO but with higher sample efficiency, as shown in Figure 2. Thus, HP3O/HP3O+ are able to achieve a desirable trade-off in practice between sample efficiency and computational time. These experiments used a local machine with an NVIDIA RTX 4090. Additional results regarding wall-clock time for diverse methods to reach a certain reward are included in Appendix A.13.

**Robustness.** We also compute the *explained variance* LaHuis et al. (2014) for all algorithms under consideration for evaluating robustness. Please check the Appendix A.7 for more details about this metric. Intuitively, it quantifies how good a model is to explain the variations in the data. Therefore, the higher the explained variance of a model, the more the model is able to explain the variations in trajectories. Essentially, the data in this work are trajectories produced by different policies, leading to a data distribution drift. Therefore, explained variance can, to some extent, be viewed as an indicator of how well an algorithm is robust against the data distribution drift. Figure 4 shows the explained variances for HP3O and PPO in the HalfCheetah environment for five different runs with different random seeds. HP3O has the highest explained variance over all runs suggesting that it is more robust against the variations of trajectories during learning. While for PPO, its explained variance can reach large negative values during training, which indicates the training instability when the trajectories vary significantly.

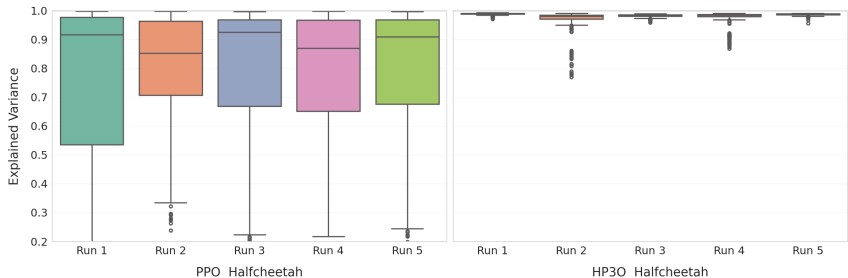

Figure 4: Explained Variance for HalfCheetah for PPO and HP3O

## 6.3 LIMITATIONS

Though theoretical and empirical results have shown that the proposed HP3O outperforms the popular baseline PPO over diverse control tasks, some limitations need to be discussed for potential improvement in the future. First, HP3O/HP3O+ require more hyperparameter tuning for the trajectory replay buffer, which can impact model performance compared to PPO. It has been acknowledged that hyperparameter tuning is critical for reinforcement learning such that for the hardest benchmarks, the already narrow basins of effective hyperparameters may become prohibitively small for our proposed algorithms, leading to poor performance. Second, in sparse reward environments, dequeuing the trajectory replay buffer can result in insufficient learning. Unlike the traditional replay buffer, which stores all experiences, our design requires the buffer to discard old trajectories so that the potential data distribution drift can be alleviated. This may cause a problem that good trajectories may only be learned once. Thus, the tradeoff between data distribution drift and learning frequency for the buffer needs to be investigated more in future work. Finally, there remains substantial room for performance improvement for the proposed algorithms compared to SAC. Further work in algorithm design is required to ensure HP3O/HP3O+ is on par with SAC but with low variance. The current ones can be regarded as one of the first steps toward bridging the gap between on-policy and off-policy methods.

## 7 CONCLUSION AND BROADER IMPACTS

In this work, we presented a novel hybrid-policy reinforcement learning algorithm by incorporating a replay buffer into the popular PPO algorithm. Specifically, we utilized random sampling to reuse samples generated by the prior policies to improve the sample efficiency of PPO. We developed HP3O and theoretically derived its policy improvement lower bound. Subsequently, we designed a new advantage function in HP3O+ and presented a modified lower bound to provide theoretical guarantees. We investigated the stationary point convergence for HP3O and used several continuous control environments and baselines to showcase the superiority of the proposed algorithms. Additionally, we focused on variance reduction while maintaining high reward returns, encouraging the community to consider both high rewards and variance reduction. The theoretical claims of higher sample efficiency and variance reduction were empirically supported.

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

# A  APPENDIX

In this section, we present additional analysis and experimental results as a supplement to the main contents. To conveniently refer to the theoretical results, we repeat the statements for all lemmas and theorems.

## A.1  ADDITIONAL THEORETICAL ANALYSIS

**Lemma 4.** *(Lemma 6.1 in Kakade & Langford (2002)) For any policies $\hat{\pi}$ and $\pi$, we have*

$$J(\hat{\pi}) - J(\pi) = \frac{1}{1-\gamma}\mathbb{E}_{s\sim d_{\hat{\pi}}}[\mathbb{E}_{a\sim\hat{\pi}(\cdot|s)}[A^{\pi}(s,a)]] \tag{7}$$

Lemma 4 signifies the cumulative return difference between two policies, $\pi$ and $\hat{\pi}$.

**Lemma 5.** *Consider any two policies $\hat{\pi}$ and $\pi$. Then the total variation distance between the state visitation distributions $d^{\hat{\pi}}$ and $d^{\pi}$ is bounded by*

$$\delta(d^{\pi}, d^{\hat{\pi}}) \leq \frac{\gamma}{1-\gamma}\mathbb{E}_{s\sim d^{\hat{\pi}}}[\delta(\pi,\hat{\pi})(s)], \tag{8}$$

*where $\delta(\pi,\hat{\pi})(s)$ is defined in Lemma 1.*

The proof follows similarly from Achiam et al. (2017). Next we present the proof for Lemma 2.

**Lemma 2:** Consider a present policy $\pi_k$, and any reference policy $\pi_r$. We then have, for any future policy $\pi$,

$$J(\pi) - J(\pi_k) \geq \frac{1}{1-\gamma}\mathbb{E}_{(s,a)\sim d^{\pi_r}}\Big[\frac{\pi(a|s)}{\pi_r(a|s)}A^{\pi_k}(s,a)\Big] - \frac{2\gamma C^{\pi}_{\pi_k}}{(1-\gamma)^2}\mathbb{E}_{s\sim d^{\pi_r}}[\delta(\pi,\pi_r)(s)], \tag{9}$$

where $C^{\pi}_{\pi_k}$ and $\delta(\pi,\pi_r)(s)$ are defined as in Lemma 1.

*Proof.* The proof is similar to the proof of Lemma 7 in Queeney et al. (2021). We start from the equality in Lemma 4 by adding and subtracting the term

$$\frac{1}{1-\gamma}\mathbb{E}_{s\sim d^{\pi_r}}[\mathbb{E}_{a\sim\pi(\cdot|s)}[A^{\pi_k}(s,a)]] \tag{10}$$

$\square$

With this, we obtain the following relationship:

$$J(\pi) - J(\pi_k) = \frac{1}{1-\gamma}\mathbb{E}_{s\sim d^{\pi_r}}[\mathbb{E}_{a\sim\pi(\cdot|s)}[A^{\pi_k}(s,a)]]$$

$$+ \frac{1}{1-\gamma}(\mathbb{E}_{s\sim d^{\pi}}[\mathbb{E}_{a\sim\pi(\cdot|s)}[A^{\pi_k}(s,a)]] - \mathbb{E}_{s\sim d^{\pi_r}}[\mathbb{E}_{a\sim\pi(\cdot|s)}[A^{\pi_k}(s,a)]])$$

$$\geq \frac{1}{1-\gamma}\mathbb{E}_{s\sim d^{\pi_r}}[\mathbb{E}_{a\sim\pi(\cdot|s)}[A^{\pi_k}(s,a)]]$$

$$- \frac{1}{1-\gamma}|\mathbb{E}_{s\sim d^{\pi}}[\mathbb{E}_{a\sim\pi(\cdot|s)}[A^{\pi_k}(s,a)]] - \mathbb{E}_{s\sim d^{\pi_r}}[\mathbb{E}_{a\sim\pi(\cdot|s)}[A^{\pi_k}(s,a)]]|$$

$$(11)$$

The last inequality follows from the Triangle inequality. Subsequently, we can bound the second term of the last inequality using Hölder's inequality:

$$\frac{1}{1-\gamma}|\mathbb{E}_{s\sim d^{\pi}}[\mathbb{E}_{a\sim\pi(\cdot|s)}[A^{\pi_k}(s,a)]] - \mathbb{E}_{s\sim d^{\pi_r}}[\mathbb{E}_{a\sim\pi(\cdot|s)}[A^{\pi_k}(s,a)]]|$$

$$\leq \frac{1}{1-\gamma}\|d^{\pi} - d^{\pi_r}\|_1\|\mathbb{E}_{a\sim\pi(\cdot|s)}[A^{\pi_k}(s,a)]\|_{\infty},$$

$$(12)$$

where $d^{\pi}$ and $d^{\pi_r}$ both signify the state visitation distributions. In light of the definition of total variation distance and Lemma 3, the following relationship can be obtained accordingly

$$\|d^{\pi} - d^{\pi_r}\|_1 = 2\delta(d^{\pi}, d^{\pi_r}) \leq \frac{2\gamma}{1-\gamma}\mathbb{E}_{s\sim d^{\pi_r}}[\delta(\pi,\pi_r)(s)]. \tag{13}$$

Also note that

$$\|\mathbb{E}_{a\sim\pi(\cdot|s)}[A^{\pi_k}(s,a)]\|_{\infty} = \max|\mathbb{E}_{a\sim\pi(\cdot|s)}[A^{\pi_k}(s,a)]| = C^{\pi}_{\pi_k}. \tag{14}$$

Hence, substituting Eq. 13 and Eq. 14 into Eq. 12 and combining Eq. 11 yields the following inequality:

$$J(\pi) - J(\pi_k) \geq \frac{1}{1-\gamma}\mathbb{E}_{s\sim d^{\pi_r}}[\mathbb{E}_{a\sim\pi(\cdot|s)}[A^{\pi_k}(s,a)]] - \frac{2\gamma C^{\pi}_{\pi_k}}{(1-\gamma)^2}\mathbb{E}_{s\sim d^{\pi_r}}[\delta(\pi,\pi_r)(s)]. \tag{15}$$

Finally, without loss of generality, we assume that the support of $\pi$ is contained in the support of $\pi_r$ for all states, which is true for common policy representations used in policy optimization. We can rewrite the first term on the right hand side of the last inequality as

$$\frac{1}{1-\gamma}\mathbb{E}_{s\sim d^{\pi_r}}[\mathbb{E}_{a\sim\pi(\cdot|s)}[A^{\pi_k}(s,a)]] = \frac{1}{1-\gamma}\mathbb{E}_{(s,a)\sim d^{\pi_r}}[\frac{\pi(a|s)}{\pi_r(a|s)}A^{\pi_k}(s,a)], \tag{16}$$

which leads to the desirable results.

**Theorem 1:** Consider prior policies $|\mathcal{B}|$ randomly sampled from the replay buffer $R$ with indices $i = 0, 1, ..., |\mathcal{B}| - 1$. For any distribution $v = [v_1, v_2, ..., v_{|\mathcal{B}|}]$ over the $|\mathcal{B}|$ prior policies, and any future policy $\pi$ generated by HP3O in Algorithm 1, the following relationship holds true

$$J(\pi) - J(\pi_k) \geq \frac{1}{1-\gamma}\mathbb{E}_{i\sim v}[\mathbb{E}_{(s,a)\sim d^{\pi_i}}[\frac{\pi(a|s)}{\pi_i(a|s)}A^{\pi_k}(s,a)]] - \frac{\gamma C^{\pi}_{\pi_k}\epsilon}{(1-\gamma)^2}, \tag{17}$$

where $C^{\pi}_{\pi_k}$ is defined as in Lemma 1.

*Proof.* Based on the definition of total variation distance, we have that

$$\mathbb{E}_{s\sim d^{\pi_k}}[\delta(\pi,\pi_k)(s)] = \mathbb{E}[\frac{1}{2}\int_{a,\mathcal{A}}|\pi(a|s) - \pi_k(a|s)|\mathrm{d}a]. \tag{18}$$

We still make the assumption that the support of $\pi$ is contained in the support of $\pi_k$ for all states, which is true for the common policy representations used in policy optimization. Then, by multiplying and dividing by $\pi_k(a|s)$, we can observe that

$$\mathbb{E}_{s\sim d^{\pi_k}}[\delta(\pi,\pi_k)(s)] = \mathbb{E}[\frac{1}{2}\int_{a,\mathcal{A}}\pi_k(a|s)|\frac{\pi(a|s)}{\pi_k(a|s)} - 1|\mathrm{d}a]$$

$$= \frac{1}{2}\mathbb{E}_{(s,a)\sim d^{\pi_k}}[|\frac{\pi(a|s)}{\pi_k(a|s)} - 1|] \leq \frac{\epsilon}{2}.$$

$$(19)$$

The last inequality follows from the setup of PPO. With prior policies $\pi_i, i = 0, 1, 2, ..., |\mathcal{B}| - 1$, we assume that the support of $\pi$ is contained in the support of $\pi_i$ for all states, which is true for common policy representations used in policy optimization. Based on Lemma 2, we can obtain

$$J(\pi) - J(\pi_k) \geq \frac{1}{1-\gamma}\mathbb{E}_{(s,a)\sim d^{\pi_i}}[\frac{\pi(a|s)}{\pi_i(a|s)}A^{\pi_k}(s,a)] - \frac{2\gamma C^{\pi}_{\pi_k}}{(1-\gamma)^2}\mathbb{E}_{s\sim d^{\pi_i}}[\delta(\pi,\pi_i)(s)]. \tag{20}$$

Consider policy weights $v = [v_1, v_2, ..., v_{|\mathcal{B}|}]$ over the policies in the minibatch $\mathcal{B}$. Thus, for any choice of distribution $v$, the convex combination determined by $v$ of the $|\mathcal{B}|$ lower bounds given by

the last inequality yields the lower bound

$$J(\pi) - J(\pi_k) \geq \frac{1}{1-\gamma}\mathbb{E}_{i \sim v}[\mathbb{E}_{(s,a) \sim d^{\pi_i}}[\frac{\pi(a|s)}{\pi_i(a|s)}A^{\pi_k}(s,a)]]$$

$$- \frac{2\gamma C^\pi_{\pi_k}}{(1-\gamma)^2}\mathbb{E}_{i \sim v}[\mathbb{E}_{s \sim d^{\pi_i}}[\delta(\pi, \pi_i)(s)]]. \tag{21}$$

Combining Eq. 19 and Eq. 21, with some mathematical manipulation, results in the desirable conclusion. Now we're ready to prove Lemma 3. □

**Lemma 3:** Consider a present policy $\pi_k$, and any reference policy $\pi_r$. We then have, for any future policy $\pi$,

$$J(\pi) - J(\pi_k) \geq \frac{1}{1-\gamma}\mathbb{E}_{(s,a) \sim d^{\pi_r}}[\frac{\pi(a|s)}{\pi_r(a|s)}\hat{A}^{\pi_k}(s,a)]$$

$$- \frac{2\gamma \hat{C}^\pi_{\pi_k}}{(1-\gamma)^2}\mathbb{E}_{s \sim d^{\pi_r}}[\delta(\pi, \pi_r)(s)] \tag{22}$$

$$- \frac{2\gamma C^{\pi_k}}{(1-\gamma)^2}\mathbb{E}_{s \sim d^{\pi_r}}[\delta(\pi, \pi_r)(s)],$$

where $\hat{C}^\pi_{\pi_k} = \max_{s \in \mathcal{S}}|\mathbb{E}_{a \sim \pi(\cdot|s)}[\hat{A}^{\pi_k}(s,a)]|$, $\delta(\pi, \pi_r)(s)$ is defined as in Lemma 1, $C^{\pi_k} = \max_{s \in \mathcal{S}}|V^{\pi_k^*}(s) - V^{\pi_k}(s)|$.

*Proof.* Due to Lemma 1, we have

$$J(\pi) - J(\pi_k) = \frac{1}{1-\gamma}\mathbb{E}_{s \sim d^\pi}[\mathbb{E}_{a \sim \pi(\cdot|s)}[A^{\pi_k}(s,a)]]$$

$$= \frac{1}{1-\gamma}\mathbb{E}_{s \sim d^\pi}[\mathbb{E}_{a \sim \pi(\cdot|s)}[Q^{\pi_k}(s,a) - V^{\pi_k}(s)]] \tag{23}$$

$$= \frac{1}{1-\gamma}\mathbb{E}_{s \sim d^\pi}[\mathbb{E}_{a \sim \pi(\cdot|s)}[Q^{\pi_k}(s,a) - V^{\pi_k^*}(s) + V^{\pi_k^*}(s) - V^{\pi_k}(s)]].$$

Let $\hat{A}^{\pi_k}(s,a) = Q^{\pi_k}(s,a) - V^{\pi_k^*}(s)$ and $G^{\pi_k}(s) = V^{\pi_k^*}(s) - V^{\pi_k}(s)$ such that

$$J(\pi) - J(\pi_k) = \frac{1}{1-\gamma}\mathbb{E}_{s \sim d^\pi}[\mathbb{E}_{a \sim \pi(\cdot|s)}[\hat{A}^{\pi_k}(s,a)]] + \frac{1}{1-\gamma}\mathbb{E}_{s \sim d^\pi}[\mathbb{E}_{a \sim \pi(\cdot|s)}[G^{\pi_k}(s)]]. \tag{24}$$

Define $\|G^{\pi_k}(s)\|_\infty = \max_{s \in \mathcal{S}}|V^{\pi_k^*}(s) - V^{\pi_k}(s)| = C^{\pi_k}$. Follow similarly the proof from Lemma 2, we can attain the relationship as follows:

$$J(\pi) - J(\pi_k) \geq \frac{1}{1-\gamma}\mathbb{E}_{s \sim d^{\pi_r}}[\mathbb{E}_{a \sim \pi(\cdot|s)}[\hat{A}^{\pi_k}(s,a)]] + \frac{1}{1-\gamma}\mathbb{E}_{s \sim d^{\pi_r}}[G^{\pi_k}(s)]$$

$$- \frac{2\gamma \hat{C}^\pi_{\pi_k}}{(1-\gamma)^2}\mathbb{E}_{s \sim d^{\pi_r}}[\delta(\pi, \pi_r)(s)] \tag{25}$$

$$- \frac{2\gamma C^{\pi_k}}{(1-\gamma)^2}\mathbb{E}_{s \sim d^{\pi_r}}[\delta(\pi, \pi_r)(s)].$$

□

The fact that $\min_{s \in \mathcal{S}}|V^{\pi_k^*}(s) - V^{\pi_k}(s)| = 0$ retains the desirable result.

**Theorem 2:** Consider prior policies $|\mathcal{B}|$ randomly sampled from the replay buffer $R$ with indices $i = 0, 1, ..., |\mathcal{B}| - 1$. For any distribution $v = [v_1, v_2, ..., v_{|\mathcal{B}|}]$ over the $|\mathcal{B}|$ prior policies, and any future policy $\pi$ generated by HP3O+ in Algorithm 1, the following relationship holds true

$$J(\pi) - J(\pi_k) \geq \frac{1}{1-\gamma}\mathbb{E}_{i \sim v}[\mathbb{E}_{(s,a) \sim d^{\pi_i}}[\frac{\pi(a|s)}{\pi_i(a|s)}\hat{A}^{\pi_k}(s,a)]] - \frac{\gamma \hat{C}^\pi_{\pi_k}\epsilon}{(1-\gamma)^2} - \frac{\gamma C^{\pi_k}\epsilon}{(1-\gamma)^2}, \tag{26}$$

where $\hat{C}^\pi_{\pi_k}$ and $C^{\pi_k}$ are defined as in Lemma 3.

*Proof.* Following the proof techniques in Theorem 1 and combining the conclusion from Lemma 3 obtains Eq. 26. □

## A.2 RISK OF OVERFITTING?

In our approach, each set of sampled trajectories includes the current best action trajectory in the buffer, but we use a uniform distribution to sample mini-batch data points from all the trajectories rather than only focusing on the best one. Additionally, the number of sampled trajectories is a tunable parameter that we adjust based on the specific environment. Therefore, we ensure that the model is exposed to a diverse set of experiences, which also helps mitigate the risk of overfitting. Another important point is that our trajectory buffer operates on a FIFO (FirstIn-First-Out) basis. As newer trajectories are added to the buffer, the oldest ones are replaced. This buffer maintains a dynamic structure where trajectories are continually updated to reflect the most recent learning and also helps to reduce distribution drift. We expect that these newer trajectories are more likely to be better-performing as they are generated from the most current learned policy. All these techniques are implemented in our buffer and help to balance exploration with prioritizing higher-performing trajectories while also reducing the risk of overfitting.

## A.3 INCORPORATION OF THE WORST TRAJECTORIES

In our approach, we prioritize leveraging higher-performing trajectories to optimize the agent's learning efficiency and to accelerate convergence toward optimal policies. This focus allows the agent to reinforce successful behaviors more effectively. However, we understand the concern regarding forgetting catastrophic behaviors, which could potentially lead to the agent's catastrophic behaviors. In practice, the FIFO buffer and uniform sampling from the sampled trajectories make sure that a diverse range of experiences, including suboptimal or catastrophic behaviors, are preserved to some extent within the buffer. This diversity helps the agent to maintain a broad understanding of the environment, including both successful and unsuccessful strategies. Additionally, while we do not explicitly prioritize the worst trajectories, our approach does not entirely discard them. By maintaining a diverse buffer, the agent is still exposed to these behaviors, which can serve as alerting examples. This exposure helps the agent learn to avoid repeating such catastrophic actions without the need to focus on the worst trajectories explicitly. We believe this balance allows the agent to focus on learning from successful strategies while still retaining an understanding of less optimal behaviors, reducing the risk of catastrophic forgetting.

## A.4 SAMPLE EFFICIENCY ANALYSIS

In this section, we present the sample efficiency analysis for the proposed HP3O algorithm, compared to the vanilla PPO algorithm, which remains the most popular on-policy scheme so far. Though the analysis is conducted particularly for the comparison between PPO and HP3O, the techniques apply extensively to other on-policy policy-gradient-based algorithms whenever they satisfy the conservative policy iteration property Kakade & Langford (2002); Achiam et al. (2017) to have the policy improvement lower bounds. In this study, we aim to show how the off-policy sample reuse significantly affects the original sample efficiency PPO has. We will not directly show the exact sample complexity of HP3O and the improvement on top of PPO. For instance, to arrive at an $\varepsilon$-optimality for policy gradient-based algorithms, a few works Zhong & Zhang (2024); Zanette et al. (2021); Dai et al. (2023); Sherman et al. (2023) have revealed the exact complexity with respect to $\varepsilon$, but only for MDPs with linear function approximation. The exact sample complexity analysis for the on-policy PPO algorithm remains extremely challenging and requires a substantial amount of non-trivial effort. Thereby, in this paper, we disclose the impact of off-policy sample reuse on the tradeoff between sample efficiency and learning stability.

To start with the comparison between PPO and HP3O, we denote by $\epsilon_H$ and $\epsilon_P$ the clipping parameters for HP3O and PPO. Such a clipping parameter indicates the worst-case expected performance loss of update at every time step. We next present a lemma that shows the relationship between $\epsilon_H$ and $\epsilon_P$.

**Lemma 6.** *Consider prior policies $|\mathcal{B}|$ randomly sampled from the replay buffer $R$ with indices $i = 0, 1, ..., |\mathcal{B}| - 1$. For any distribution $v = [v_1, v_2, ..., v_{|\mathcal{B}|}]$ over the $|\mathcal{B}|$ prior policies, both HP3O and PPO have the same worst-case expected performance loss at every update when the clipping parameters satisfy the following condition:*

$$\epsilon_H = \frac{\epsilon_P}{\mathbb{E}_{i \sim v}[i + 1]}. \tag{27}$$

*Proof.* Recall from PPO such that

$$\frac{2\gamma C^\pi_{\pi_k}}{(1-\gamma)^2}\mathbb{E}_{s\sim d^{\pi_k}}[\delta(\pi,\pi_k)(s)] \leq \frac{2\gamma C^\pi_{\pi_k}}{(1-\gamma)^2}\frac{\epsilon_P}{2}. \tag{28}$$

For HP3O, its penalty term in the policy improvement lower bound in Theorem 1 can be upper bounded by using the Triangle inequality. Therefore, we have the following relationship

$$\frac{2\gamma C^\pi_{\pi_k}}{(1-\gamma)^2}\mathbb{E}_{i\sim v}[\mathbb{E}_{s\sim d^{\pi_i}}[\delta(\pi,\pi_i)(s)]]$$

$$\leq \frac{2\gamma C^\pi_{\pi_k}}{(1-\gamma)^2}\mathbb{E}_{i\sim v}\left[\sum_{j=0}^{i}\mathbb{E}_{s\sim d^{\pi_i}}[\delta(\pi_{j+1},\pi_j)(s)]\right]. \tag{29}$$

The last inequality holds if the prior policies are in a chronological order based on their histories. In practice, we do not set such an order for them, but due to the FIFO strategy we have leveraged, they can still be set in this for the sake of analysis. Since we still resort to the clipping mechanism in HP3O, each policy update approximately bounds each expected total variation distance $\mathbb{E}_{s\sim d^{\pi_i}}[\delta(\pi_{j+1},\pi_j)(s)]$ by $\frac{\epsilon_H}{2}$, which follows analogously from that in PPO. With this in hand, we are now ale to further bound Eq. 29 in the following relationship

$$\frac{2\gamma C^\pi_{\pi_k}}{(1-\gamma)^2}\mathbb{E}_{i\sim v}[\mathbb{E}_{s\sim d^{\pi_i}}[\delta(\pi,\pi_i)(s)]]$$

$$\leq \frac{2\gamma C^\pi_{\pi_k}}{(1-\gamma)^2}\mathbb{E}_{i\sim v}\left[\frac{\epsilon_H}{2}(i+1)\right] \tag{30}$$

$$\leq \frac{2\gamma C^\pi_{\pi_k}}{(1-\gamma)^2}\frac{\epsilon_H}{2}\mathbb{E}_{i\sim v}[i+1]$$

Comparing the bounds in Eq. 28 and Eq. 30 yields the desirable result. $\square$

Lemma 6 technically shows us that if the two clipping parameters $\epsilon_H$ and $\epsilon_P$ satisfy the condition of $\epsilon_H = \frac{\epsilon_P}{\mathbb{E}_{i\sim v}[i+1]}$, the worst-case expected performance loss at each update remains roughly the same. This intuitively makes sense as HP3O leverages prior policies from the replay buffer to update the policy model, which requires it to perform smaller updates. A benefit from this is to make policy updates more frequently, thus schematically stabilizing policy learning. In what follows, we present more analysis about this tradeoff.

To ease the analysis, we assume that the policies in the training batch $\mathcal{B}$ are randomly sampled with uniform policy weights, i.e., $v_i = \frac{1}{|\mathcal{B}|}$, for $i = 0, 1, ..., |\mathcal{B}| - 1$, for collecting data to train the network models. However, more advanced techniques such as Prioritized Experience Replay (PER) Schaul et al. (2015) can be applied accordingly. In each episode, we also assume that for PPO, it requires $N = Mn$ samples for sufficiently training the critic and actor networks, where $M$ is the number of mini-batch and $n$ is the batch size. In this setting, PPO makes one episodic update upon the current policy $\pi_k$ by traversing $N$ samples generated by $\pi_k$. However, for HP3O, since there exist multiple policies prior to $\pi_k$, it is able to make $M$ updates sourced from different prior policies per $N$ samples collected from $\mathcal{B}$, as long as $|\mathcal{B}| \leq M$. Thus, we next show that HP3O is able to increase the change in the total variational distance of the policy throughout training, without sacrificing stability, when it is compared to PPO.

**Theorem 3.** *Suppose that $|\mathcal{B}| = M$ and that the policies in the training batch $\mathcal{B}$ are randomly sampled with uniform policy weights, i.e., $v_i = \frac{1}{|\mathcal{B}|}$, for $i = 0, 1, ..., |\mathcal{B}| - 1$. Then, HP3O has a larger frequency of change in total variation distance of the policy throughout training by a factor of $\frac{2M}{M+1}$ compared to PPO, while using the same number of samples for each update as PPO.*

*Proof.* Pertaining to Lemma 6 and the fact that $|\mathcal{B}| = M$, we have the following relationship:

$$\epsilon_H = \frac{\epsilon_P}{\frac{1}{M}\sum_{i=0}^{M-1}(i+1)} = \frac{2\epsilon_P}{M+1}. \tag{31}$$

PPO makes one episodic policy update after $N$ samples are collected, say from $k$ to $k+1$, which yields a policy change of $\frac{\epsilon_P}{2}$ in terms of the total variation distance. While for HP3O, it resorts to data from prior policies to obtain $N$ samples and makes $M$ policy updates, as mentioned before. This results in the overall policy change of

$$M\frac{\epsilon_H}{2} = \frac{2M}{M+1}\frac{\epsilon_P}{2}. \tag{32}$$

Thus, HP3O has a larger frequency of changes in the total variation distance of the policy throughout training by a factor of $\frac{2M}{M+1}$ compared to PPO, with the same number of samples. □

By far, we have discussed the tradeoff between learning stability and sample size that biases toward learning stability when maintaining the same sample size as in PPO. Alternatively, we can perceive the problem from another perspective, in which HP3O needs to increase the sample size while maintaining the same change in total variation distance throughout training. A formal result is summarized as follows.

**Theorem 4.** *Suppose that $|\mathcal{B}| = 2M - 1$ and that the policies in the training batch $\mathcal{B}$ are randomly sampled with uniform policy weights, i.e., $v_i = \frac{1}{|\mathcal{B}|}$, for $i = 0, 1, ..., |\mathcal{B}| - 1$. Thus, HP3O increases the sample size used for each policy update by a factor of $\frac{2M-1}{M}$ compared to PPO, simultaneously maintaining the same change in the total variation of distance of the policy throughout training as PPO.*

*Proof.* As $|\mathcal{B}| = 2M - 1$, HP3O uses $(2M - 1)n$ to calculate each policy update from the prior to the new policy, compared to $Mn$ samples used in PPO. Hence, HP3O increases the sample size used for each policy update by a factor of $\frac{2M-1}{M}$ compared to PPO. Immediately, based on Lemma 6, we can obtain

$$\epsilon_H = \frac{\epsilon_P}{\frac{1}{2M-1} \sum_{i=0}^{2M-2}(i+1)} = \frac{\epsilon_P}{M}. \tag{33}$$

We have shown in Theorem 3 that PPO makes one policy update with $N$ samples collected, while HP3O makes $M$ policy updates with the same number of samples collected. We then have

$$\frac{M\epsilon_H}{2} = \frac{\epsilon_P}{2}. \tag{34}$$

This implies that the overall change in total variation distance in HP3O is the same as in PPO. □

One implication from Theorem 3 and 4 is that HP3O with uniform policy weights enhances the tradeoff between learning stability and sample efficiency in the vanilla PPO when $|\mathcal{B}|$ is selected between $[M, 2M - 1]$. This also motivates us to set the FIFO strategy as the selected training batch $\mathcal{B}$ cannot deviate too far away from the current policy. Otherwise, the negative impact of distribution drift could be extreme.

## A.5   HP3O vs. HP3O+

In the last subsection, we have shown that HP3O enables more frequent changes in the total variational distance of the policy throughout training, with the smaller updates. Though more changes in the total variational distance of the policy may help improve the sample efficiency, but in order to address the distribution drift, smaller updates are the resulting outcome, possibly slowing down the convergence. Hence, introducing the best trajectory $\pi^*$ in HP3O+ assists in mitigating this issue. Since it can increase the update, while maintaining the same number of changes as in HP3O. Such a behavior is empirically shown to enhance the model performance. We summarize the larger update in the total variational distance in a formal theoretical result as follows.

**Theorem 5.** *Denote by $D_{TV}^H$ and $D_{TV}^{H+}$ the updates of total variational distance of the policies for HP3O and HP3O+, respectively, at the time step $k$. Then we have $D_{TV}^H \leq D_{TV}^{H+}$ for all $k$.*

*Proof.* In light of Theorem 1 and Theorem 2, we know that $D_{TV}^H = \frac{\gamma C_{\pi_k}^\pi \epsilon}{(1-\gamma)^2}$ and $D_{TV}^{H+} = \frac{\gamma \hat{C}_{\pi_k}^\pi \epsilon}{(1-\gamma)^2} + \frac{\gamma C^{\pi_k} \epsilon}{(1-\gamma)^2}$. We next show the the latter is bounded below by the former. As $A^{\pi_k}(s,a) = \hat{A}^{\pi_k}(s,a) + G^{\pi_k}(s)$, $\hat{A}^{\pi_k}(s,a) = Q^{\pi_k}(s,a) - V^{\pi_k^*}(s)$, and $G^{\pi_k}(s) = V^{\pi_k^*}(s) - V^{\pi_k}(s)$, we have the following relationship

$$A^{\pi_k}(s,a) = \hat{A}^{\pi_k}(s,a) + V^{\pi_k^*}(s) - V^{\pi_k}(s) \tag{35}$$

Taking the expectation of the action $a \sim \pi(\cdot|s)$ on both sides yields

$$\mathbb{E}_{a \sim \pi(\cdot|s)}[A^{\pi_k}(s,a)] = \mathbb{E}_{a \sim \pi(\cdot|s)}[\hat{A}^{\pi_k}(s,a)] + V^{\pi_k^*}(s) - V^{\pi_k}(s), \tag{36}$$

which leads to

$$|\mathbb{E}_{a \sim \pi(\cdot|s)}[A^{\pi_k}(s,a)]| = |\mathbb{E}_{a \sim \pi(\cdot|s)}[\hat{A}^{\pi_k}(s,a)] + V^{\pi_k^*}(s) - V^{\pi_k}(s)|$$
$$\leq |\mathbb{E}_{a \sim \pi(\cdot|s)}[\hat{A}^{\pi_k}(s,a)]| + |V^{\pi_k^*}(s) - V^{\pi_k}(s)|. \tag{37}$$

This last inequality is due to the Triangle inequality. Taking the maximum operator of the state $s \sim \mathcal{S}$ on both sides results in the following:

$$\max_{s \sim \mathcal{S}} |\mathbb{E}_{a \sim \pi(\cdot|s)}[A^{\pi_k}(s,a)]| \leq \max_{s \sim \mathcal{S}} |\mathbb{E}_{a \sim \pi(\cdot|s)}[\hat{A}^{\pi_k}(s,a)]| + \max_{s \sim \mathcal{S}} |V^{\pi_k^*}(s) - V^{\pi_k}(s)|. \tag{38}$$

Multiplying both sides in the above inequality by $\frac{\gamma\epsilon}{(1-\gamma)^2}$ yields the desirable result. □

## A.6 TRAINING RESULTS FOR OTHER ENVIRONMENTS

The following plot in Figure 5a presents the training curves obtained by training both the baseline algorithms and our policy. These results further support our claim in the main paper that our policy reduces variance while maintaining a high reward at the end.

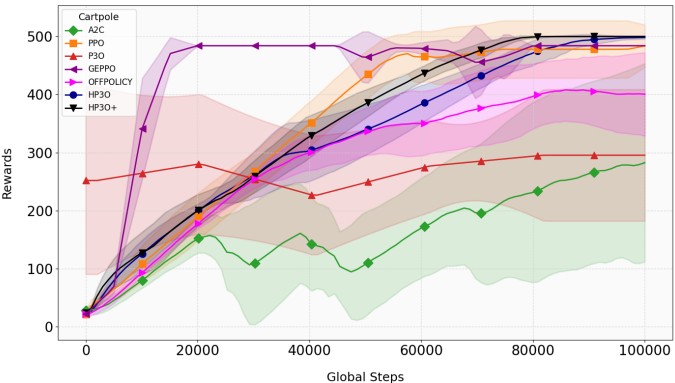

(a) Training performance of HP3O and PPO on the Cartpole environment over 100k steps.

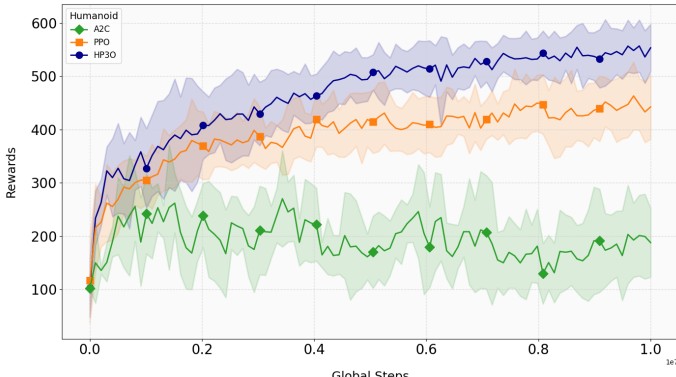

(b) Training performance of HP3O and PPO on the Humanoid environment over 10 million steps.

Figure 5: Comparison of HP3O and PPO training curves across different environments. (a) shows the performance on Cartpole, while (b) shows the performance on Humanoid.

## A.7 ADDITIONAL EXPERIMENTAL RESULTS

**Definition of explained variance.** The explained variance (EV) measures the proportion to which a mathematical model accounts for the variation of a given data set, which can be mathematically

defined in the following:

$$EV = 1 - \frac{Var(y - \hat{y})}{Var(y)}, \tag{39}$$

where $y$ is the groundtruth and $\hat{y}$ is the prediction. EV values typically vary from 0 to 1. In some scenarios, the value may be a large negative number, which indicates a poor prediction of $y$. Explained variance is a well-known metric in reinforcement learning, particularly for assessing the accuracy of value function predictions. In our experiment, explained variance was used to evaluate how well the value function predicts actual returns. The different runs correspond to separate training instances with different random seeds. The explained variance score is a risk metric that measures the dispersion of errors in a dataset. A score closer to 1.0 is better, as it indicates smaller squares of standard deviations of errors.

## A.8 EXPLAINED VARIANCE FOR OTHER ENVIRONMENTS

Explained variance is a well-known metric in reinforcement learning, particularly for assessing the accuracy of value function predictions. In our experiment, explained variance was used to evaluate how well the value function predicts actual returns. The different runs correspond to separate training instances with different random seeds.

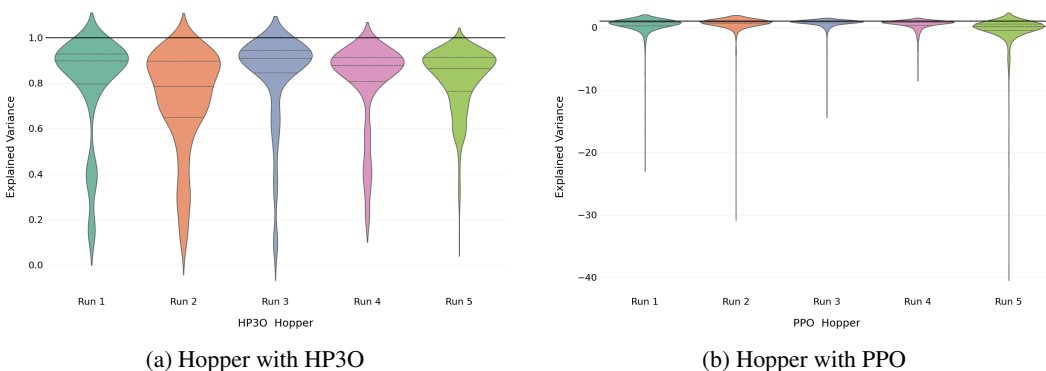

(a) Hopper with HP3O        (b) Hopper with PPO

Figure 6: Explained Variance for Hopper

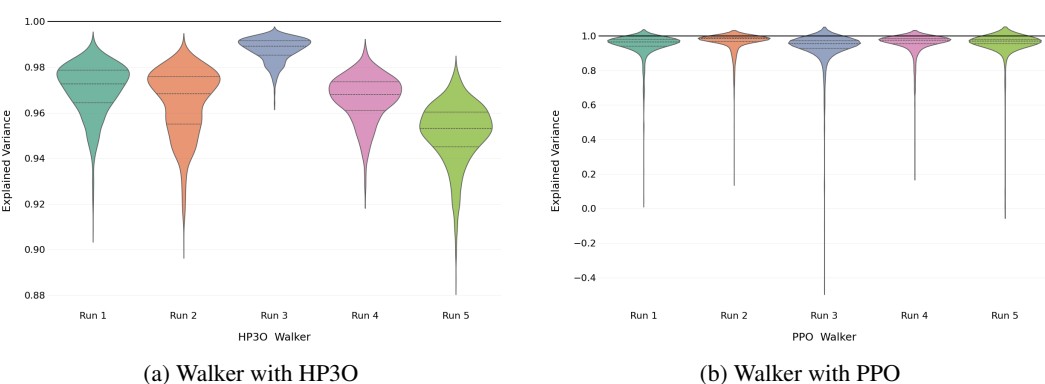

(a) Walker with HP3O        (b) Walker with PPO

Figure 7: Explained Variance for Walker

## A.9 SAC TRAINING BENCHMARKS

The following plots showcase the benchmark training results obtained by using the SAC policy. In some environments, SAC shows a relatively large variance. A notable disadvantage of SAC is that it only works with continuous action spaces.

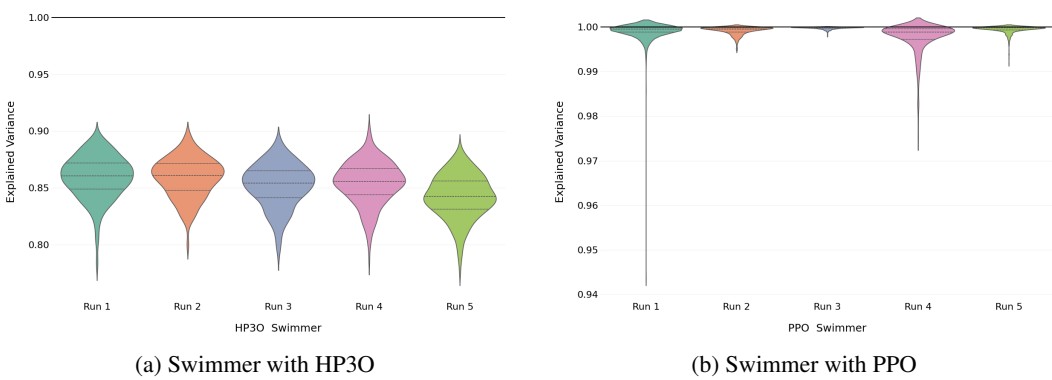

(a) Swimmer with HP3O        (b) Swimmer with PPO

Figure 8: Explained Variance for Swimmer

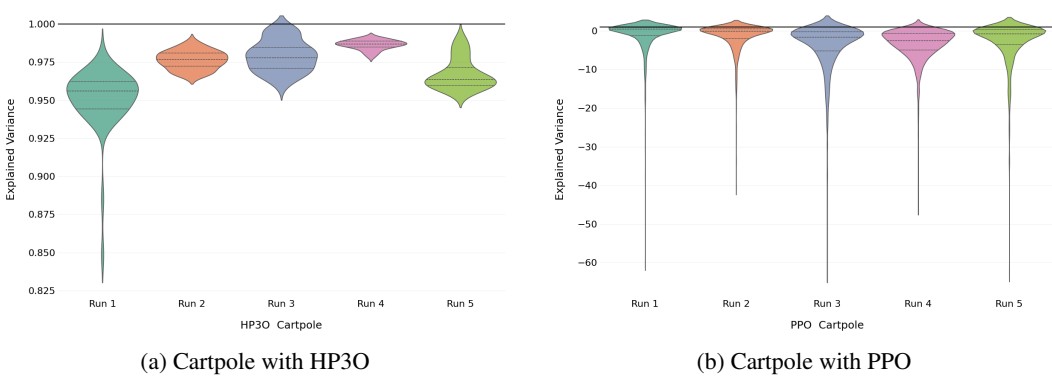

(a) Cartpole with HP3O        (b) Cartpole with PPO

Figure 9: Explained Variance for Cartpole

### A.10 EXPERIMENTAL CONFIGURATION

Experiments were performed on a local machine with an Intel Core i7-14700 CPU, 128 GB of RAM, and NVIDIA RTX 4090 GPU. We provide detailed information on our algorithms' hyperparameters for all environments in our GitHub repository, which will be released once the paper is published.

### A.11 DATA STORAGE FOR RL TRAINING

- **Trajectory Buffer**: Stores complete trajectories $\tau$ as sequences of $(s_t, a_t, r_t, s_{t+1})$.
    - **Data Types**: Arrays of states, actions, rewards, and next states.
    - **Dimensions**:
        * States $s_t$: Typically $\mathbb{R}^n$ where $n$ is the dimension of the state space.
        * Actions $a_t$: Depends on the action space, usually $\mathbb{R}^m$ where $m$ is the dimension of the action space.
        * Rewards $r_t$: Scalar values.
        * Next states $s_{t+1}$: Same as states $s_t$.

### A.12 CONVERGED REWARD

In this subsection, we present the converged reward at the last time step. In some environments, the training curves do not fully converge, which may make it challenging to assess the ultimate performance. However, for consistency across all algorithms, we maintained the same number of training time steps for each experiment. This allows for a fair comparison of sample efficiency across different methods, even if the algorithms did not always fully converge within the given time frame.

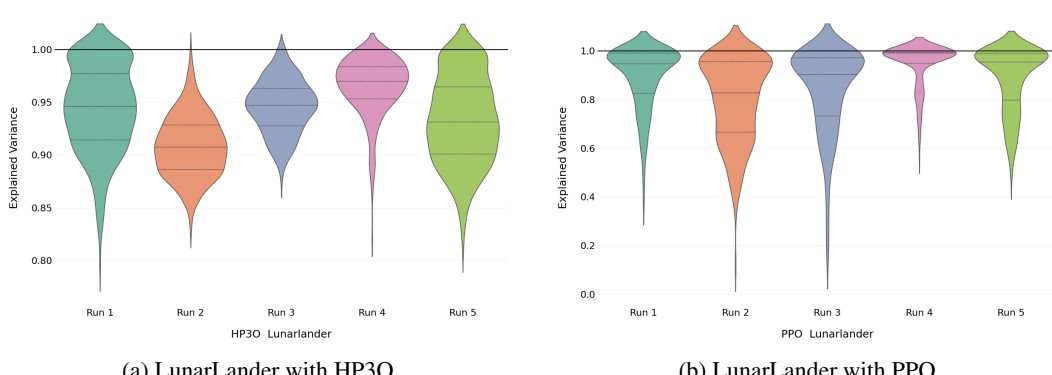

(a) LunarLander with HP3O

(b) LunarLander with PPO

Figure 10: Explained Variance for LunarLander

Additionally, in some environments, we do present converged training curves, demonstrating the capabilities of the algorithms. In the reinforcement learning community, it is a common practice to show learning curves at a fixed number of steps for comparative analysis, even if full convergence is not always achieved. Notably, papers on SAC Haarnoja et al. (2018), PPO Schulman et al. (2017), GEPPO Queeney et al. (2021), and Off-Policy PPO Meng et al. (2023) follow similar practices, with many of the environments presented in these works employing non-converged curves to provide valuable insights into training dynamics and sample efficiency. Table 2 shows the detailed converged reward performance of all different RL algorithms over different continuous tasks. In order to ensure a fair comparison between GEPPO and our method, we first analyzed the performance differences between the PPO baselines in our implementation vs. in the GEPPO repository. These discrepancies were primarily due to variations in implementation details (e.g., leveraging TensorFlow packages, early version of Mujoco environment), which significantly impacted the baseline performance. To address the discrepancies, we normalized the PPO baseline results to match our implementation. As a result, both baselines produced comparable outcomes. We then applied the same normalization factor to the GEPPO results, repeating this procedure for each environment to ensure fair and consistent comparisons across all settings.

Table 2: Summary of mean and standard deviation of rewards for each policy across diverse environments (in the form Mean $\pm$ Std) at or close to the converged stage. The bold one represents the best reward performance.

| Environment | A2C | GEPPO | HP3O | HP3O+ | OffPolicy | P3O | PPO |
|---|---|---|---|---|---|---|---|
| CartPole | $282.47 \pm 170.87$ | $21.76 \pm 2.54$ | $498.25 \pm 2.51$ | $\mathbf{500.00 \pm 0.00}$ | $400.31 \pm 72.60$ | $295.38 \pm 113.44$ | $483.59 \pm 36.70$ |
| Halfcheetah | $334.16 \pm 302.14$ | $2156.31 \pm 1024.06$ | $3523.20 \pm 565.39$ | $\mathbf{3967.47 \pm 244.80}$ | $738.11 \pm 274.89$ | $1251.17 \pm 517.84$ | $2276.87 \pm 902.20$ |
| Hopper | $120.31 \pm 48.29$ | $976.33 \pm 68.36$ | $988.67 \pm 16.53$ | $\mathbf{1891.35 \pm 79.47}$ | $961.64 \pm 76.69$ | $1107.49 \pm 281.71$ | $946.90 \pm 64.86$ |
| InvertedPendulum | $71.26 \pm 76.20$ | $463.02 \pm 0.00$ | $956.66 \pm 36.20$ | $\mathbf{1000.00 \pm 0.00}$ | $11.24 \pm 3.22$ | $810.77 \pm 46.26$ | $463.02 \pm 445.28$ |
| LunarLander | $-69.13 \pm 105.52$ | $136.07 \pm 12.33$ | $\mathbf{225.91 \pm 10.97}$ | $146.63 \pm 7.10$ | $114.68 \pm 106.51$ | $-624.72 \pm 292.51$ | $130.58 \pm 16.53$ |
| Swimmer | $12.99 \pm 6.25$ | $133.83 \pm 6.69$ | $340.00 \pm 4.18$ | $\mathbf{343.40 \pm 1.41}$ | $157.73 \pm 107.07$ | $32.75 \pm 14.32$ | $131.98 \pm 38.76$ |
| Walker | $134.05 \pm 51.54$ | $1140.31 \pm 389.75$ | $1934.91 \pm 152.69$ | $1895.29 \pm 98.74$ | $\mathbf{2093.24 \pm 370.08}$ | $1777.91 \pm 465.01$ | $1150.36 \pm 97.18$ |

## A.13 COMPUTATIONAL EFFICIENCY

To probe particularly the computational efficiency of diverse algorithms presented in this study, we compare them in the wall-clock time spent to reach a certain reward in the HalfCheetah environment. Due to the time limitation, we are unable to obtain results for all other environments, while including them in the final version. Such an investigation offers us useful insights about which methods are more practically feasible and deployable given the limited real-time budget. Figure 12 shows the specific performance of wall-clock time cost for different approaches reaching the rewards of 1100 and 2100, respectively. One immediate observation from the results is that GEPPO requires significantly more time to converge compared to all other schemes. For a couple of algorithms, such as A2C and OffPolicy, the training progresses are pretty slow, eventually failing to achieve rewards of 1100 and 2100 in the HalfCheetah environment. Another implication of interest from the results is that at the beginning, PPO may progress faster, compared to both HP3O and HP3O+. However, due to its on-policy behavior, the sample inefficiency issue still affects the overall training progress. Different

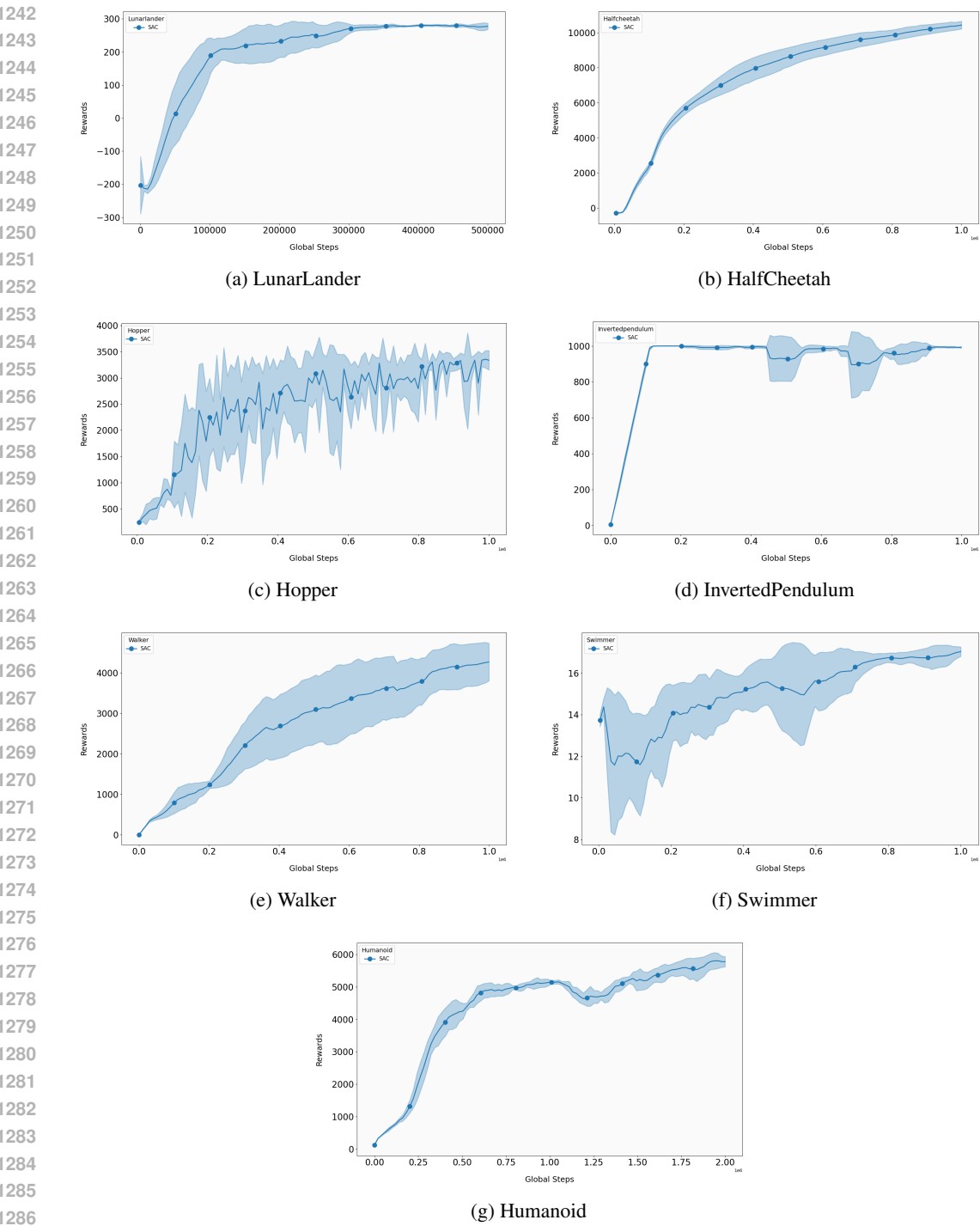

Figure 11: Training curves for SAC across various environments. The solid curves indicate the mean, while the shaded areas represent the standard deviation over the five runs.

from that, both HP3O and HP3O+ make consistent progress throughout the training process and take minimal time to achieve certain rewards. Between them, HP3O+ has slightly better performance, which empirically validates our conclusion from Theorem 5. The finding also complies with that in Figure 3b.

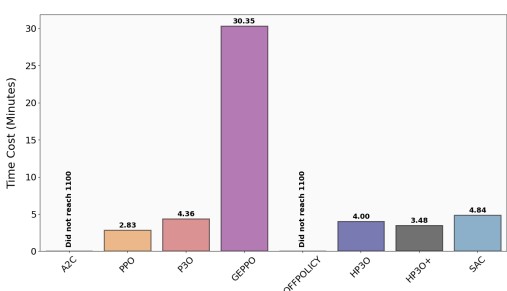 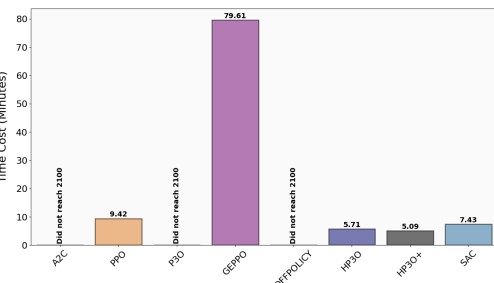

(a) Time to reach the reward of 1100 for HalfCheetah       (b) Time to reach the reward of 2100 for HalfCheetah

Figure 12: Time taken by algorithms to reach specific reward thresholds in the HalfCheetah environment.

## A.14  IMPACT OF BUFFER SIZE AND MEMORY USAGE

In this subsection, we also discuss the impact of the buffer size on the model performance qualitatively. The other two aspects included here are the discussion on the sampling strategy and the trade-off in sparse reward settings.

- **Large Trajectory Buffer Size**: A larger trajectory buffer size allows us to store a greater number of diverse trajectories. This diversity can enhance generalization and reduce overfitting, as the agent learns from a broad range of experiences. However, in sparse reward settings, maintaining a large trajectory buffer may mean that the inclusion of outdated or less relevant trajectories could introduce instability and slow down learning, as the agent may be exposed to experiences that no longer align with its current policy.

- **Small Trajectory Buffer Size**: A smaller trajectory buffer retains fewer trajectories, which typically results in the agent learning from more recent experiences that are closely aligned with the current policy. This can improve stability, as updates are based on recent, relevant data. However, a smaller buffer can reduce the diversity of sampled experiences, leading to an increased risk of overfitting and limiting the agent's ability to effectively explore different parts of the environment.

- **Sampling Strategy**: Our sampling strategy also plays a critical role in managing the trade-off between stability and performance. By ensuring that the best return trajectory is always included in the sampled trajectories, we provide a strong guiding signal that improves policy performance. The sample rate, where we evenly sample from each selected trajectory, helps in maintaining a balance between exploration and exploitation, as well as in utilizing high-quality trajectories effectively.

- **Trade-off in Sparse Reward Settings**: In sparse reward environments, the need to maintain high-quality trajectories becomes even more crucial. A large trajectory buffer can help capture rare, valuable experiences, but the risk of dequeuing these valuable trajectories before they can contribute meaningfully to learning is higher. Ensuring that the best trajectory is always sampled helps mitigate this issue, but the buffer size still influences how effectively these rare rewards are retained and leveraged.

Regarding memory consumption, the total memory usage depends on the following:

1. **Number of Trajectories in the Buffer (Buffer Size)**: We use a fixed-size replay buffer, which holds $N$ trajectories.

2. **Average Length of Trajectories** ($T$): Since trajectories have varying lengths, the memory usage will depend on the average trajectory length.

3. **Dimensionality of State and Action Spaces**: Let $d_s$ be the dimension of the state, and $d_a$ be the dimension of the action. Each step in a trajectory stores both state and action information.

4. **Data type size**: Denote by $n$ the specific data type size.

The memory consumption $(M)$ can be roughly estimated as:
$$M = NTn(d_s + d_a)$$
For example, if $N = 1000$ trajectories are stored, each with an average length of 200 steps, and assuming $d_s = 17$, $d_a = 6$, and using 32-bit floats (4 bytes), the memory requirement would be:
$$M = 1000 \times 200 \times (17 + 6) \times 4 \text{ bytes} = 18,400,000 \text{ bytes} \approx 18.4 \text{ MB}$$
This calculation provides an estimate, but the actual memory usage may vary depending on the environment and the specific implementation of the replay buffer.

### A.15 EVALUATION RESULTS

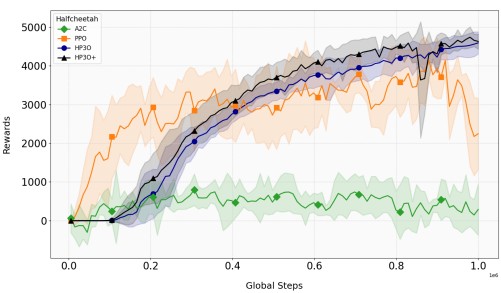
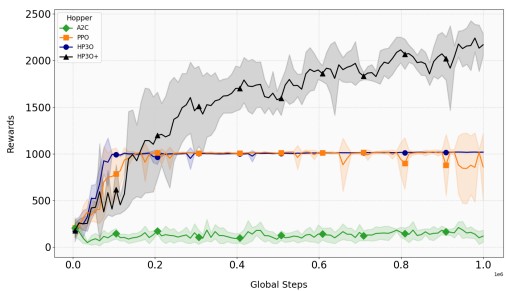

(a) Evaluation plot for the HalfCheetah environment.     (b) Evaluation plot for the Hopper environment.

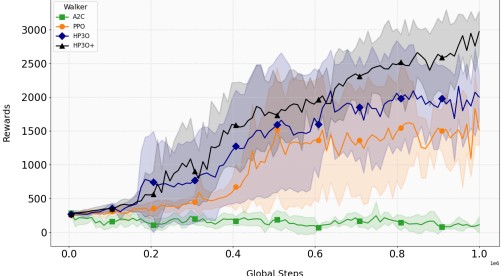

(c) Evaluation plot for the Walker environment.

Figure 13: Evaluation plots for the HalfCheetah, Hopper, and Walker environments during the evaluation stage every 5000 steps. Each experiment includes five different runs with various random seeds. The solid curves indicate the mean, while the shaded areas represent the standard deviation over the five runs.

The evaluation results align with our training expectations. Overall, our presented models, HP3O and HP3O+, consistently outperform the baseline models across all environments, achieving higher rewards while maintaining relatively low variance. The PPO baseline performs well initially but tends to be less sample efficient and has relatively higher variance, whereas A2C struggles to reach comparable performance.

The results clearly demonstrate that our presented models, HP3O and HP3O+, are better equipped for these environments. This also verifies our claim from the training analysis. Both HP3O and HP3O+ combine improved sample efficiency and reduced variance, leading to more stable learning outcomes and higher cumulative rewards. These advantages enable our models to not only outperform the baselines but also maintain robustness and efficiency across diverse environments. Due to the time limitation, we are unable to obtain results for other environments with other methods, and will include additional results in the final version.

