# OpenReview forum: "HP3O: Hybrid-Policy Proximal Policy Optimization with Best Trajectory"
_ICLR.cc/2025/Conference — Submitted to ICLR 2025_

### Official Review · Reviewer_Ye97 · 2024-10-22

**Soundness:** 2
**Presentation:** 3
**Contribution:** 2
**Rating:** 6
**Confidence:** 4

**Summary:**

This paper proposes a new hybrid-policy RL method called HP3O for continuous control tasks, which employs a trajectory replay buffer to reduce variance  and improve sample efficiency. HP3O is simple but effective with theoretical guarantees on policy improvement. The experiments on Mujoco control tasks demonstrate its effectiveness empirically. This paper is well written and the method is clearly described.

**Strengths:**

1. The new method HP3O(+) is simple and easy to implement. The effectiveness of HP3O policy improvement is analyzed theoretically in detail.
2. The PPO is widely used in applications. Thus, this method can be utilized to improve the performance of downstream tasks such as robotics control tasks.
3. The paper is well-written with clear descriptions of the method and the experiments.

**Weaknesses:**

The main concerns lie in the experiments:

1. More experiment results with (more) baseline methods are expected. In this paper, authors compare HP3O(+) with PPO, A2C, P3O, and SAC. However, some results are not shown in the current version:

   1. The learning curves of SAC & HP3O+ in Fig. 2 are absent. It is recommended that Fig. 2 and Fig. 3 can be merged, with more results of HP3O+ in more experimental tasks.

   2. Similarly, HP3O+ in absent in Fig. 4(a), which is important to demonstrate the performance of HP3O+ on variance.

   3. More baseline methods are expected, such as on-policy methods constructed based on PPO. Besides, comparisons with off-policy methods such as SAC are also meaningful. As illustrated in the paper, HP3O achieves superior sample efficiency compared to SAC, which is normal and acceptable.
2. This paper claims that SAC achieves higher sample efficiency at the cost of runtime complexity. Fig. 4(b) gives the training times of methods in *HalfCheetah* with 1M steps. However, it is more reasonable to give the training times of each method with *same episode returns*, which is more meaningful in practical applications. Besides, more results in other tasks are expected to demonstrate this claim. It is recommended that the curve of episode return - training time is clearer than Fig. 4(b).

3. Some latex errors \ref in the manuscripts.

**Questions:**

1. Does the experiment utilize parallel gymnasium environment settings, or is each method trained with single environment? In practical applications, parallel environments are widely used such as ISAAC simulator with thousands  of robots trained simultaneously, which reduce training times with higher computation costs. Does this method still achieve similar higher performance than other methods?
2. In this paper, the author claims that ``A notable disadvantage of SAC is that it only works with continuous action spaces. " However, SAC can be utilized in discrete action spaces [1]. In this paper, HP3O(+) is implemented in continuous action spaces. Is this method effective in discrete action space?
3. The experiments are conducted in several Mujoco continuous control tasks. Is this method effective in locomotion tasks with higher action spaces, such as Humanoid-v2 task?

[1] Christodoulou P. Soft actor-critic for discrete action settings[J]. arXiv preprint arXiv:1910.07207, 2019.

---

> ### Author Response · Authors · 2024-11-20
> **Response to Reviewer Ye97's comments**
>
> **Weakness 1:**
>
> We appreciate your suggestions for adding additional experiments to improve the results.
>
> $•$ Learning Curves of SAC and HP3O+ in Figure 2:
>
> We are in the process of generating these results and will include them in an updated version of the manuscript. The reason we initially did not include SAC in Figure 2 is that our primary focus was on comparing HP3O+ with the PPO variant and other on-policy methods.
>
> Additionally, we will merge Figures 2 and 3 to present a comprehensive comparison, and the additional results for HP3O+ across all environments will be included in Figure 2 as well.
>
> $•$ HP3O+ in Figure 4(a):
>
> We are currently updating Figure 4(a) to include the additional results for HP3O+. The figure will be adjusted accordingly to reflect these new results.
>
>
> $•$ More Baseline Methods:
>
> We appreciate your suggestion to include additional baselines in our work. We are actively working on incorporating more baseline results in the revised version of the paper. As Reviewer 2 suggested, we will present some of the useful baseline algorithms in our manuscript.
>
> We have faced challenges in implementing certain baselines due to significant differences in experimental setups. Specifically, some of the baseline algorithms were originally implemented in very different settings of the environments, which require extensive code refactoring. As a result, we were unable to include some of these baselines in our experiments. However, we are committed to adding as many relevant baselines as possible, as suggested by the reviewers.
>
> We will update the manuscript accordingly with these additional experimental results, including the additional plots as suggested by the reviewer.
>
> **Weakness 2:**
>
> Thank you for providing such an interesting idea to plot the curve of episode return against training time. We are currently reproducing such plots by measuring the training time for each method until they reach similar episode returns. Additionally, we will include supplementary tables showing the time usage based on the same return to provide a clearer illustration of the comparison between methods. We will update our manuscript with these additional plots and tables.
>
> **Weakness 3:**
>
> Thank you for pointing out the reference errors in the manuscript. We apologize for those issues. We will review the manuscript and correct all errors.
>
> **Question 1:**
>
> Thank you for your question regarding the experimental setup and the use of parallel environments. Our training setup can utilize multiple environments in parallel. In our code, this is easily configurable by adjusting the number of environment parameters, which allows using multiple environments for parallel training.
>
> The reason we conducted our experiments with a single environment setting is to certify a controlled fair comparison across different algorithms, which also minimizes variability that can arise due to differences in the number of parallel environments. However, we acknowledge that parallel environments are widely used in practical applications to speed up training. Our proposed HP3O method is designed to be capable of leveraging parallelization. We leave this as an open choice for users based on real-world hardware restrictions. The parallelization setup in HP3O follows a similar implementation manner as other methods, such as PPO in Stable Baselines3.
>
> We will clarify in the manuscript that our approach is available for parallel environment settings. Thank you again for highlighting this important aspect.
>
> **Question 2:**
>
> We apologize for the incorrect statement regarding SAC. The phrase should be changed from "only works" to "primarily works" with continuous action spaces. Thank you for pointing out the availability of versions of SAC that can handle discrete action spaces.
>
> In our work, we primarily focus on extending PPO, and the experiments have largely been conducted in continuous action spaces. This choice is due to the fact that continuous environments generally present more challenges to RL agents. Since the HP3O structure fundamentally belongs to a variant of PPO, it is expected to perform well in discrete environments as well. We have also implemented HP3O in a discrete action space environment, such as the CartPole environment. The results are presented in Figure 22 of the appendix and demonstrate that HP3O performs well in discrete settings, showing performance outperformance to PPO.
>
> We will update the manuscript to clarify this distinction and correct the statement about SAC. Thank you again for your insightful comment.

---

> ### Author Response · Authors · 2024-11-20
> **Response to Reviewer Ye97's comments (Continued)**
>
> **Question 3:**
> Thank you for your question regarding environments with higher action spaces.
>
> We have conducted experiments in the Humanoid-v2 environment. Preliminary results show that our HP3O method is more effective compared to baseline PPO and other on-policy methods, as claimed in our manuscript.
>
> We will include these results in our manuscript and provide an evaluation across all environments with this high-dimensional humanoid environment. We appreciate your patience and your insightful question.

---

> > ### Comment · Reviewer_Ye97 · 2024-11-25
> > **Response to the rebuttal**
> >
> > Thanks for the detailed response, which solves some concerns. We appreciate the effort of authors on this work.
> >
> > However, some concerns still exist. Some key experiment results are still absent. For example,
> > 1. More experiments with other baseline methods are expected, such as previous off-policy PPO methods.
> > 2. "Episode return - time cost" curves are expected, especially for HP3O+, to demonstrate the effectiveness and efficiency in practical applications.
> > 3. Experiments on HP3O+ are insufficient. More results are needed to demonstrate the effectiveness of HP3O+, OR clarify the application scenarios for HP3O+ compared to other methods.
> >
> > Above all, there still exist some key issues in the current version, which are needed to demonstrate the effectiveness of this work. Further discussions with more necessary experiment results are expected.  Thus, I keep my rating (5).

---

> ### Author Response · Authors · 2024-11-27
> **Response to Reviewer Ye97's comments**
>
> Thank you once again for your valuable feedback. We have addressed the concerns raised as follows:
>
> 1. We have added the other baseline results, including both off-policy PPO and Geppo methods.
>
> 2. In the supplementary materials, we have included the "Episode Return - Time Cost" curves. From these plots, we observe that both presented models demonstrate greater effectiveness and efficiency. We have discussed these results further in the supplementary section.
>
> 3. We have conducted additional experiments for HP3O+, and the revised manuscript now includes these results for all presented environments.
>
> We appreciate all your comments. Please feel free to reach out with any further questions or concerns, and we will do our best to address them.

---

> > ### Comment · Reviewer_Ye97 · 2024-12-02
> >
> > Thanks for your response. The results of the new methods are excellent compared to previous methods, both on training efficiency and time costs. According to the revision of the manuscript, I have updated the rating (5 to 6).
> >
> > Besides, it is still recommended that the authors give more results on the training time in more environments, similar to Fig. 12 in Appendix. This is important in practice. Because the manuscript is not allowed to be revised currently, these results may be  added in the final version.

---

> > > ### Author Response · Authors · 2024-12-02
> > > **Response to Reviewer Ye97's comments**
> > >
> > > Dear Reviewer Ye97,
> > >
> > > Thank you for your thoughtful feedback and for updating the rating. We appreciate your recognition of our new method's performance.
> > >
> > > Regarding your recommendation to include additional results on training time across more environments, we are committed to completing experiments in all environments and will incorporate these results into the final version.
> > >
> > > We are grateful for all your constructive suggestions. Please let us know if there are any further comments or clarifications needed.
> > >
> > > Best regards,
> > >
> > > The authors of Paper 11700

---

### Official Review · Reviewer_iV5K · 2024-11-04

**Soundness:** 3
**Presentation:** 3
**Contribution:** 2
**Rating:** 6
**Confidence:** 3

**Summary:**

This paper introduces Hybrid-Policy PPO (HP3O), a variant of Proximal Policy Optimization (PPO) designed to improve sample efficiency and reduce high variance. HP3O integrates on-policy and off-policy learning through a replay buffer that stores recent trajectories, using a “first in, first out” (FIFO) strategy to retain only the most recent experiences. By incorporating the trajectory with the highest return along with randomly sampled trajectories from the buffer for policy updates, HP3O empirically reduces variance while preserving a policy improvement guarantee. The authors provide rigorous proofs for the policy improvement lower bounds of both HP3O and its variant, HP3O+. Empirical evaluations across continuous control environments demonstrate the effectiveness of HP3O compared to other baselines.

**Strengths:**

1. The hybrid-policy approach of HP3O is an innovative extension of PPO, blending on-policy stability with off-policy sample efficiency by storing recent trajectories in a replay buffer and selectively sampling high-reward experiences.
2. The paper is rigorous both in theory and empirical validation. It presents a clear theoretical framework, including policy improvement bounds that account for the hybrid on-policy/off-policy nature of HP3O.
3. The paper is well-structured and provides detailed explanations of key concepts.

**Weaknesses:**

1. While the paper incorporates the replay buffer concept from off-policy reinforcement learning into PPO, the scope of this structure modification shows limited originality.
2. The training curves in Figures 1 and 2 do not appear to have fully converged, making it difficult to assess the ultimate performance that each model could achieve.
3. The impact of the replay buffer size has not been discussed theoretically or evaluated empirically, leaving an important aspect of the method unexplored.

**Questions:**

1. In Algorithm 1 Line 16, should $G_t−V_\phi(s_t)$ be replaced by $\hat{A}_t^{\pi_k}$ to be consistent with HP3O+?
2. Could you provide the final, converged reward for each algorithm in the experiments to better demonstrate their performance?
3. Could you include a discussion on the impact of replay buffer size, either through theoretical analysis or additional simulation results?

---

> ### Author Response · Authors · 2024-11-20
> **Response to Reviewer iV5K's comments**
>
> **Weakness 1:**
>
> Thank you for your feedback.
>
> We acknowledge that the use of replay buffers in reinforcement learning is a concept derived from off-policy methods. However, the primary contribution of our work lies in adapting and integrating this concept into the on-policy setting of PPO, which presents several unique challenges. Specifically, the integration required careful handling of the balance between on-policy and off-policy data, addressing issues such as maintaining the benefits of PPO's policy gradient updates while leveraging experience replay to stabilize training and reduce sample variance.
>
> Our approach, HP3O, uses a novel trajectory buffer that retains high-quality trajectories, which enhances the exploration-exploitation balance without compromising PPO's core stability. The key innovation lies in:
>
> $•$ The use of trajectory prioritization to selectively store and replay the best experiences. Notably, when selecting data for updates, we always include the trajectory with the best return, combined with additional random samples from the rest of the buffer. This makes sure that high-value experiences are consistently utilized while still maintaining diversity through random sampling.
>
> $•$ The design of the hybrid on-off policy update mechanism, which adapts PPO's policy update rules to work effectively with the replayed trajectories, mitigating common issues such as policy divergence or instability due to outdated data.
>
> $•$ The incorporation of a variance reduction mechanism that works synergistically with the trajectory replay buffer, ensuring that the additional replay does not introduce harmful biases and instead enhances the training process.
>
> Our approach uses optimal experiences to guide policy learning by always including the best trajectory. As a result, we demonstrate improved performance and sample efficiency while maintaining the theoretical guarantees of on-policy updates. We appreciate your constructive feedback and hope that this addresses your concerns. We will update our manuscript accordingly to include these changes.
>
> **Weakness 2:**
>
> Thank you for your suggestions regarding the convergence of the training curves. We believe the reviewer intended to refer to Figures 2 and 3.
>
> We agree that in some environments, the training curves do not fully converge, which may make it challenging to assess the ultimate performance. However, for consistency across all algorithms, we maintained the same number of training time steps for each experiment. This allows for a fair comparison of sample efficiency across different methods, even if the algorithms did not always fully converge within the given time frame. Additionally, in some environments, we do present converged training curves, demonstrating the capabilities of the algorithms.
>
> In the reinforcement learning community, it is a common practice to show learning curves at a fixed number of steps for comparative analysis, even if full convergence is not always achieved. Notably, papers on SAC [1], PPO [2], GePPO [3], and Off-Policy PPO [4] follow similar practices, with many of the environments presented in these works employing non-converged curves to provide valuable insights into training dynamics and sample efficiency.
>
> We will update the manuscript to clarify our approach and the reasoning behind presenting non-converged training curves. Thank you again for your valuable feedback.
>
> [1] Haarnoja, T., Zhou, A., Abbeel, P., \& Levine, S. (2018, July). Soft actor-critic: Off-policy maximum entropy deep reinforcement learning with a stochastic actor. In International conference on machine learning (pp. 1861-1870). PMLR.
>
> [2] Schulman, J., Wolski, F., Dhariwal, P., Radford, A., \& Klimov, O. (2017). Proximal policy optimization algorithms. arXiv preprint arXiv:1707.06347.
>
> [3] Queeney, J., Paschalidis, Y., \& Cassandras, C. G. (2021). Generalized proximal policy optimization with sample reuse. Advances in Neural Information Processing Systems, 34, 11909-11919.
>
> [4] Meng, W., Zheng, Q., Pan, G., \& Yin, Y. (2023, June). Off-policy proximal policy optimization. In Proceedings of the AAAI Conference on Artificial Intelligence (Vol. 37, No. 8, pp. 9162-9170).

---

> ### Author Response · Authors · 2024-11-20
> **Response to Reviewer iV5K's comments (Continued)**
>
> **Weakness 3:**
>
> We have conducted some offline investigations on the effect of replay buffer size, taking into account computational time and memory usage. These offline evaluations have helped us understand practical trade-offs between maintaining a larger buffer for more diverse sampling and managing computational and memory costs.
>
> The key factor that impacts training is the batch $\mathcal{B}$ selected from the replay buffer. The size of the buffer plays a crucial role in determining the diversity and recency of the experiences that make up each training batch. A larger replay buffer may increase the diversity of experiences but also requires careful management to avoid incorporating outdated trajectories that could reduce policy stability. Conversely, a smaller replay buffer improves recency but limits the diversity of training data, which may hinder generalization.
>
> We plan to conduct more extensive empirical evaluations on the impact of replay buffer size in future work and include those results. In the meantime, we will expand our discussion in the manuscript to include these practical considerations, highlighting the key role of the replay buffer in balancing performance and stability. In the final version, we will include some ablation studies in the draft.
>
>
> **Question 1:**
>
> You are correct $G_t$ - $V_{\phi}(s_t)$ should be replaced by $\hat{A}^{\pi_t}_t$ for consistency with the HP3O+ formulation. Line 16 in Algorithm 1 is used for both HP3O and HP3O+.
>
> We will correct this in the revised version of the manuscript. Thank you again for your careful review and for bringing this to our attention.
>
> **Question 2:**
>
> There are several reasons why we did not provide fully converged reward plots for all environments. Our primary goal was to conduct a fair comparison by using the same number of training steps for all algorithms across environments, which is a common practice in the RL community. For more details, please refer to our response to Weakness 2.
>
> **Question 3:**
>
> Thank you for your question regarding the impact of replay buffer size.
>
> There are several factors that influenced our decision not to provide detailed experimental results on the impact of replay buffer size at this stage. Due to computational limitations and resource constraints, conducting such experiments requires a large amount of training time and resources. Currently, we are proceeding with further analysis, and for additional context, please refer to our response to Weakness 3.

---

> ### Author Response · Authors · 2024-11-27
> **Response to Reviewer iV5K's comments (Continued)**
>
> We have uploaded a revised version of our work based on all the provided feedback. A detailed summary of these updates is included in the revision. Please let us know if you have any further questions or need additional clarification. Thank you.

---

> > ### Comment · Reviewer_iV5K · 2024-12-02
> >
> > Thank you for your response. I appreciate your detailed explanation of how your approach balances exploration and exploitation while maintaining stability. Your response on the training curve convergence and the replay buffer size are clear and solid, which solved my questions. Overall, your clarifications have resolved my concerns, thank you again for your efforts in addressing the feedback.

---

> > > ### Author Response · Authors · 2024-12-02
> > > **Response to Reviewer iV5K's comments**
> > >
> > > Dear Reviewer iV5K,
> > >
> > > Thank you very much for your kind response. I sincerely appreciate your acknowledgment of our efforts to address the points you raised.
> > >
> > > If it is possible, I would like to kindly ask whether you might consider revisiting your score, as we have worked carefully to resolve the issues you raised. It would be truly crucial for us, and your support would mean a lot.
> > >
> > > Thank you again for your valuable time, insights, and consideration. Your feedback has been immensely helpful in strengthening our work.
> > >
> > > Best regards,
> > >
> > > The authors of Paper 11700

---

### Official Review · Reviewer_zRpn · 2024-11-04

**Soundness:** 2
**Presentation:** 3
**Contribution:** 2
**Rating:** 6
**Confidence:** 3

**Summary:**

This paper proposes Hybrid-Policy Proximal Policy Optimization (HP3O) and a variant HP3O+, which incorporate a replay buffer with standard PPO, to improve the sample efficiency while mitigating the high variance common in off-policy algorithms. They derived the policy improvement lower bound for both HP3O and HP3O+. Experiments conducted on continuous control environments demonstrate that the proposed algorithms perform comparably or better than baselines, including PPO, A2C [Peng et al. 2018] and P3O [Chen et al. 2023].

**Strengths:**

- the proposed approach combines the strengths of on-policy and off-policy methods
- offers theoretical insights
- validated the efficacy in continuous control experiments

**Weaknesses:**

1. The baselines in the experiments are limited: six baseline methods are listed in Table 1 as related works but only one of them is used as a baseline in the experiments. The paper explains that other methods were excluded because “corresponding code bases are either inaccessible or problematic”. However,  at least three of these methods have publicly available codebases, including the most relevant work GePPO.  It would be beneficial to clarify the specific issues that prevents their inclusion:

  - GePPO (official): https://github.com/jqueeney/geppo
  - Policy-on-off PPO (official): https://github.com/rasoolfa/P3O
  - Off-policy PPO (unofficial): https://github.com/Ladun/OffPolicy-PPO

    Given that GePPO and Off-policy PPO are both hybrid methods with theoretical guarantees,  comparing with them would better illustrate the benefits of using a trajectory buffer.

2. Since HP3O+ is a proposed algorithm, its learning curve should ideally be included in Fig. 2. It would strengthen the paper if the authors add the results or elaborate why they’re unnecessary.

3. It seems that the learning curves only report performance of the training episodes. If so, I think that it’s better to report the evaluation performance instead, as is done in SAC.

4. In the limitation section, it is mentioned that dequeuing replay buffer may cause issues in sparse reward setting. On a related note, conducting ablation studies on the buffer size in general settings would provide insights into its impact on performance and stability.

Minor:
- For future work, it would be useful to explore the performance of HP3O in discrete action spaces, such as Atari, as done in PPO.
- Typo on line 282, “the the”

**Questions:**

- If the best trajectory is not added to the replay buffer, which part of the theoretical results no longer holds?
- In Theorem 2, there’s a factor of 2 in the last two terms on the right-hand side. Should these factors be canceled out instead?

---

> ### Author Response · Authors · 2024-11-20
> **Response to Reviewer zRpn's comments**
>
> **Weakness 1:**
>
> Thank you for highlighting this concern regarding the inclusion of more relevant baseline methods.
>
>  $•$ GePPO: Initially, we did not present GePPO as a baseline due to a mismatch between the Gymnasium and Mujoco environments used in our setup and those required by the original GePPO implementation. Both environments involve some subtle physical parameter differences, which could lead to unfair comparisons. However, we have now implemented GePPO using its original code version, and we will add the results to our manuscript for a more comprehensive comparison.
>
>  $•$ Policy-on-off PPO: We have attempted to implement this baseline; however, we encountered challenges due to inconsistencies between the environments used. In our current study, we have implemented all proposed algorithms in Mujoco and Gymnasium environments to ensure a fair comparison across tasks. However, the original P3O codebase is implemented specifically for the Atari game environment, with a code structure primarily designed for that context. Adapting it to work in Mujoco environments would require extensive refactoring and modification of the original implementation, which was not feasible within the scope of our current work.
>
>  $•$ Off-policy PPO: Thank you for pointing out this recent work. We have now implemented this baseline as well and will include the corresponding results in our manuscript.
>
> We appreciate your valuable suggestions and will update the manuscript accordingly.
>
>
> **Weakness 2:**
>
> The learning curve for HP3O+ was initially not included in Figure 2 due to computational resource constraints. HP3O+ requires extensive parameter fine-tuning, which is computationally intensive, and scaling these experiments across all environments required considerable resources. At the time, we believed it would be sufficient to present results for two representative environments, as shown in Figure 3, to demonstrate the variance reduction and improved stability of HP3O+ compared to other algorithms.
>
> However, we recognize the importance of providing a more comprehensive comparison. We have now conducted the necessary experiments for the remaining environments and included the learning curve for HP3O+ in Figure 2. These additions aim to provide a fuller picture of the performance improvements brought by HP3O+.
>
> We will add these additional results in the revised version of the manuscript. We hope that these additions strengthen the paper and provide a more complete understanding of the proposed algorithm's capabilities.
>
> **Weakness 3:**
> Thank you for pointing this out.
>
> We are currently in the process of implementing the evaluation results for the environment, following the same evaluation protocol used for SAC. Once completed, we will update the manuscript to include these evaluation results. We will also add an additional figure and revise the relevant discussion in the manuscript to reflect these new evaluation analyses.
>
> Thank you again for your suggestion.

---

> ### Author Response · Authors · 2024-11-20
> **Response to Reviewer zRpn's comments (Continued)**
>
> **Weakness 4:**
>
> Thank you for your insightful suggestion regarding ablation studies on the buffer size.
>
> We will expand our discussion in the limitations section to provide a more thorough analysis of how buffer size impacts performance and stability.
>
>  $•$ Large Trajectory Buffer Size: A larger trajectory buffer size allows us to store a greater number of diverse trajectories. This diversity can enhance generalization and reduce overfitting, as the agent learns from a broad range of experiences. However, in sparse reward settings, maintaining a large trajectory buffer may mean that the inclusion of outdated or less relevant trajectories could introduce instability and slow down learning, as the agent may be exposed to experiences that no longer align with its current policy.
>
>  $•$ Small Trajectory Buffer Size: A smaller trajectory buffer retains fewer trajectories, which typically results in the agent learning from more recent experiences that are closely aligned with the current policy. This can improve stability, as updates are based on recent, relevant data. However, a smaller buffer can reduce the diversity of sampled experiences, leading to an increased risk of overfitting and limiting the agent's ability to effectively explore different parts of the environment.
>
>  $•$ Sampling Strategy: Our sampling strategy also plays a critical role in managing the trade-off between stability and performance. By ensuring that the best return trajectory is always included in the sampled trajectories, we provide a strong guiding signal that improves policy performance. The sample rate, where we evenly sample from each selected trajectory, helps in maintaining a balance between exploration and exploitation, as well as in utilizing high-quality trajectories effectively.
>
>  $•$ Trade-off in Sparse Reward Settings: In sparse reward environments, the need to maintain high-quality trajectories becomes even more crucial. A large trajectory buffer can help capture rare, valuable experiences, but the risk of dequeuing these valuable trajectories before they can contribute meaningfully to learning is higher. Ensuring that the best trajectory is always sampled helps mitigate this issue, but the buffer size still influences how effectively these rare rewards are retained and leveraged.
>
> We hope this expanded discussion provides a clearer understanding of the effects of buffer size on both performance and stability, especially within the context of our trajectory buffer approach. We will continue to explore the impact of buffer size experimentally next and appreciate your suggestion for including ablation studies to further demonstrate these effects. In the final version, we will include some ablation studies in the draft.
>
> **Minor:**
>
> $•$ Exploration of HP3O and HP3O+ in discrete action spaces, such as Atari games, is an interesting direction for future work. We will ensure that this is added to the future work section in our draft.
>
> $•$ We will correct this minor typo in the revised draft. Thanks for pointing this out.
>
>
> **Question 1:**
>
> If the best trajectory is not added to the replay buffer, then HP3O+ cannot be effectively implemented as intended. Consequently, Lemma 3 and Theorem 2 will no longer hold, as these theoretical results are derived based on the HP3O+ framework, which relies on the inclusion of the best trajectory in the replay buffer. The addition of the best trajectory is fundamental to ensuring that the theoretical guarantees of variance reduction and policy improvement, as described in Lemma 3 and Theorem 2, are maintained.
>
> We will clarify this point further in the revised manuscript to emphasize the necessity of including the best trajectory for the theoretical foundations to hold. Thank you again for your valuable question.
>
> **Question 2:**
>
> Thank you for pointing out this minor mistake in Theorem 2.
> We really appreciate your careful review. Indeed, the factor of 2 should be canceled out, as the total variation distance is upper bounded by half of the clipping parameter. We will make the necessary revisions to the draft to correct this error.
> Thank you again for your insightful feedback.

---

> > ### Comment · Reviewer_zRpn · 2024-11-25
> >
> > I appreciate the authors’ detailed rebuttal. The clarifications provided are clear. Their plan to incorporate additional experimental results is promising; however, I will maintain my current score until a revised version of the paper is submitted for review.
> >
> > Regarding the theoretical results for HP3O+, I have a follow-up question. While Theorem 2 establishes a policy improvement lower bound, it remains unclear how the new lower bound compares to that of HP3O. Is there any proof that this lower bound improves over the one provided by HP3O? Such a comparison could convincingly demonstrate the advantage of HP3O+.
> >
> > Furthermore, in practice, estimating the "best value" $V^{\pi^*}(s)$ based on a single trajectory seems prone to issues when there's stochasticity in either the environment transitions or the policy itself. Could the authors comment on the implications?

---

> > > ### Author Response · Authors · 2024-11-27
> > > **Response to Reviewer zRpn's comments**
> > >
> > > Thank you for your feedback and insightful questions.
> > >
> > > We have added Theorem 5 in subsection A.5 to address your question about the policy improvement lower bound. Specifically, Theorem 5 provides a comparative analysis between the lower bounds of HP3O and HP3O+. We believe this addition clarifies how the new bound offers an advantage over the original HP3O, thereby demonstrating the theoretical improvement introduced by HP3O+.
> > >
> > > Regarding your concern about estimating the "best value" from a single trajectory in stochastic environments, we understand the challenges posed by the randomness in environment transitions and policy behaviors. To address this, we have implemented a sampling strategy that averages multiple trajectories. This approach helps reduce variance caused by stochasticity, leading to a more stable and reliable evaluation of policy performance. Subsection A.14 has been updated to provide a detailed explanation of this method.
> > >
> > > We appreciate your insightful comments and hope that the revisions address your concerns effectively. Please let us know if there are any additional points you'd like us to clarify or expand upon.

---

> > > > ### Comment · Reviewer_zRpn · 2024-12-02
> > > >
> > > > Thank you for the clarifications and I apologize for the delay in submitting my comments.
> > > >
> > > > Regarding Theorem 5, I am not sure if it shows any advantage of the lower bound of HP3O+. Specifically, $-D^H_{TV} \geq -D^{H^+}_{TV}$, combined with the observation that the first term on the RHS of Eq. (4) seems to be larger than that of Eq. (6), would yield a lower (worse) lower bound for HP3O+. Hence I'm not convinced that the theoretical results adequately explains the empirical advantage of HP3O+.
> > > >
> > > > That said, I appreciate the additional empirical results, which in my view, have strengthened the paper. I have increased my score accordingly.

---

> > > > > ### Author Response · Authors · 2024-12-03
> > > > > **Response to Reviewer zRpn's comments**
> > > > >
> > > > > Dear Reviewer zRpn,
> > > > >
> > > > > Thank you for your follow-up questions and for your positive feedback on the additional empirical results. We also really appreciate your insightful comments and the increased score for our work. Below, we provide further clarifications regarding Theorem 5 to address your concerns.
> > > > >
> > > > > Regarding your concern about Theorem 5 and the lower bound comparison, we understand your point that the inequality $-D^H_{TV} \geq -D^{H^+}_{TV}$, alongside the apparent difference in the first term on the RHS of Eq. (4) versus Eq. (6), may suggest a potentially weaker bound for HP3O+. However, it is important to note that our goal with Theorem 5 is to demonstrate the conditions under which HP3O+ can effectively balance the trade-off between exploration and stability.
> > > > > In Theorem 5, our focus is primarily on analyzing the updates brought by the total variation distance (TVD) for HP3O and HP3O+, rather than comparing the overall lower bound. As noted in the revisions in Section A.4 and in line with the analysis presented in GEPPO, HP3O facilitates more frequent but smaller changes in the TVD of the policy throughout training compared to vanilla PPO. While these frequent changes in TVD can potentially enhance sample efficiency, the smaller update size may contribute to slower convergence due to distribution drift.
> > > > >
> > > > > To address this, HP3O+ introduces a best-trajectory mechanism to increase the magnitude of updates in the TVD, while maintaining the same frequency of changes as HP3O. We understand your concern that this approach could theoretically yield a weaker lower bound for HP3O+. However, it is important to emphasize that our analysis pertains only to TVD, meaning the policy improvement lower bound for HP3O+ could still be higher despite the differences in TVD. Additionally, we would like to clarify that our theoretical analysis does not capture all practical effects, such as exploration-exploitation dynamics, which often have a significant impact on empirical performance.
> > > > >
> > > > > In particular, the interplay between the terms $-D^H_{TV}$ and $-D^{H^+}_{TV}$ is influenced by the overall configuration of the training process hyperparameter settings, which controls the structure of exploration and other related dynamics. The HP3O+ mechanism is designed to introduce a more structured exploration, contributing to the improved empirical performance of HP3O+ even if, theoretically, the bounds appear to be looser. We argue that the empirical results illustrate that this added structure often outweighs any disadvantage introduced by the theoretical lower bound, thereby leading to a net benefit in practice.
> > > > >
> > > > > To provide further context, our empirical results indicate that the HP3O+ method's ability to introduce larger updates in TVD helps in overcoming the drawbacks of smaller, incremental changes seen in HP3O, ultimately leading to a faster convergence rate. The interplay between update magnitude and frequency is crucial in balancing exploration and exploitation, and we hypothesize that this mechanism contributes to the observed empirical advantage of HP3O+ over HP3O. We will ensure that the final revision includes a more detailed discussion in the appendix to better align the theoretical findings with the empirical observations.
> > > > >
> > > > > In summary, we have carefully evaluated our theoretical results with PPO as a standard, focusing on the magnitude and frequency of TVD updates. Our empirical findings support the hypothesis that HP3O+ strikes a beneficial balance between exploration and stability, ultimately resulting in stronger empirical performance despite the potential for a looser theoretical bound. We appreciate your comments, and we will contribute to making these connections clearer in the paper.
> > > > >
> > > > > We hope this additional explanation helps address your concerns. Please let us know if there are any further questions or if you need additional clarification.
> > > > >
> > > > > Thank you once again for your constructive feedback and support.
> > > > >
> > > > > Best regards,
> > > > >
> > > > > The authors of Paper 11700

---

### Official Review · Reviewer_XcPF · 2024-11-14

**Soundness:** 2
**Presentation:** 2
**Contribution:** 2
**Rating:** 5
**Confidence:** 4

**Summary:**

This paper focuses on the issues of high variance and high sample complexity in proximal policy optimization (PPO) algorithm. To mitigate the issues, an approach of Hybrid-Policy Proximal Policy Optimization (HP3O) is proposed by utilizing a trajectory replay buffer with a "first in, first out" (FIFO) strategy. Further, another variant termed HP3O+ is proposed by leveraging a new baseline to enhance the model performance. Theoretical analysis is also provided to show the policy improvement lower bounds for the proposed algorithms. Finally, experiments were conducted on continuous control environments from Gymnasium (Walker, HalfCheetah, Lunarlander, Swimmer, Hopper, and InvertedPendulum), comparing HP3O/HP3O+ with PPO, A2C, and P3O.

**Strengths:**

- High variance in on-policy policy gradient methods, such as PPO, remains a challenge. This work aims to mitigate this issue by proposing a trajectory replay buffer with FIFO stategy so that both on-policy and off-policy experience can be utilized to balance the trade-off between sample efficiency and training stability.
- Both theoretical analysis and empirical evaluation on continuous control environments are provided to show its effectiveness.

**Weaknesses:**

- The contribution of this paper looks a little bit incremental to me. The idea of using FIFO strategy for replay buffer is quite straightforward, this is OK, but I don't see why and how this will make a big difference. It's more interesting to compare and analyze FIFO with other strategies, such as prioritized replay buffer, and show why FIFO could work well or much better than the other ones. From the theoretical perspective, what's the improvement of sample complexity over other policy gradient methods? From the empirical evaluation, the current results are insufficient to support the claim. Although the paper acknowledged that fully off-policy method, such as SAC, could work better, it's still necessary to compare the proposed methods with both the state-of-the-art methods and off-policy methods (TD3, SAC, etc.). Currently, HP3O showed the advantage over PPO, A2C, P3O on some domains, but why it can make success and why it fails, it's not clear yet. For the ablation study, it's more expected to see the proposed method with or without FIFO replay buffer and best trajectory.
- There are some missing cross-references, such as "Appendix ?? and ??".

**Questions:**

- See my comments in the weaknesses.
- How to define or measure the best trajectory in the replay buffer?
- How to implement the proposed replay buffer (with FIFO) but withoug padding? How much memory it would consume?
- Why the variance of HalfCheetach becomes larger when training goes longer?
- How do you calculate the explained variance? what are the inputs and groundtruth?

---

> ### Author Response · Authors · 2024-11-20
> **Response to Reviewer XcPF's comments**
>
> **Weakness 1:**
>
> We acknowledge that the use of the FIFO replay buffer might initially appear incremental. However, we would like to emphasize that this strategy plays a crucial role in significantly mitigating data distribution drift, which is a persistent challenge in off-policy learning. The FIFO trajectory buffer only maintains the recent trajectories, which ensures that the data used for policy updates remains representative of the current policy. This directly addresses one of the primary limitations of conventional off-policy methods, which is the divergence between past experiences and the current policy.
> Furthermore, we also leverage a trajectory buffer instead of storing isolated data points, which preserves the temporal coherence of sequences of actions and observations. This coherent data structure facilitates more effective policy updates by keeping the context intact, which reduces the high variance commonly associated with on-policy algorithms like PPO. One of the key issues with PPO is its high variance, which arises from the need to collect fresh data at each update step. Our approach blends the strengths of on-policy stability with the enhanced sample efficiency of off-policy learning in a principled manner, which we believe constitutes a meaningful advancement over existing methods.
>
> The proposed HP3O algorithm and its variant have resorted to data randomly sampled from multiple policies in a training batch that is prior to the current policy for the policy update. Thus, there exist multiple updates compared to the vanilla PPO, which only makes one policy update from the current policy to the next policy.
> In this study, we aim to show how the off-policy sample reuse significantly affects the original sample efficiency PPO has. Though direct sample policy improvement can be significantly beneficial to providing a solid theoretical foundation for the proposed algorithms, a thorough investigation is out of the scope of this study. For instance, to arrive at an $\varepsilon$-optimality for on-policy policy gradient-based algorithms, a few existing works have revealed the exact complexity with respect to $\varepsilon$, but only for MDPs with linear function approximation. The exact sample complexity analysis for the off-policy PPO algorithm with nonlinear function approximation still remains extremely challenging and requires a substantial amount of non-trivial efforts. Thereby, in this paper, we instead disclose the impact of off-policy sample reuse on the tradeoff between sample efficiency and learning stability. We will add a remark on this in the main contents and detailed analysis in the Appendix.
>
> We want to clarify that our primary focus is not on direct comparisons with fully off-policy methods, as our approach still maintains an on-policy nature despite incorporating a trajectory buffer. The trajectory buffer modifies the rollout mechanism to improve sample efficiency, but the overall structure and training process remain aligned with the on-policy characteristics of PPO. This means that while we use a replay buffer, we are fundamentally different from fully off-policy algorithms like SAC and TD3, and thus, our method should be viewed as a hybrid rather than a direct competitor to those approaches.
>
> As mentioned in Remark 1 of the manuscript, we have acknowledged that incorporating PER into our replay buffer is a potential direction for future work. The current implementation of HP3O(+) uses a FIFO strategy for managing the replay buffer, which aligns well with our goal of maintaining a balance between recent experiences and sample diversity without introducing additional computational overhead.
>
> We recognize that methods like DDPG, SAC, and TD3 are effectively able to leverage PER, prioritizing experiences based on their temporal-difference (TD) error. However, integrating PER into our approach requires careful adaptation to ensure stability within the hybrid on-off policy structure of HP3O(+), particularly given the potential biases that could arise from using prioritized samples in an on-policy context. This was beyond the scope of the current study but is a highly promising extension that we plan to explore in future work.
>
> **Weakness 2:**
>
> Thank you for pointing out the missing cross-references. We will correct all cross-referencing issues throughout our paper. We sincerely appreciate your careful review.

---

> ### Author Response · Authors · 2024-11-20
> **Response to Reviewer XcPF's comments  (Continued)**
>
> **Question 1:**
>
> Thank you for pointing out this question.
>
> In our paper, at the end of the first paragraph in Section 4, we will include an equation that defines the calculation of the best trajectory and provide a brief description. Specifically, the best trajectory in the replay buffer is defined as the one that achieves the highest cumulative reward. For each trajectory in the buffer, we compute the return as the discounted sum of rewards, and the trajectory with the highest return is selected as the best trajectory ($\tau^*$).
>
> We will further refine the manuscript to provide additional clarity regarding this definition. Thank you again for highlighting this point.
>
>
> **Question 2:**
>
> **Implementation Without Padding**
>
> In our replay buffer implementation, we handle trajectories without padding to preserve the temporal characteristics and natural variations in trajectory lengths across different environments. To achieve this:
>
> $• $ Each trajectory is stored in the replay buffer with its actual length, rather than zero-padding to match other trajectories.
>
> $• $ The replay buffer stores each trajectory as a separate entry, while maintaining its original state-action pairs.
>
> $• $ During minibatch sampling, we simply sample full trajectories, and no padding is applied since the model processes each trajectory in its entirety without introducing artificial data points.
>
>
> This approach is beneficial for environments where the trajectory lengths vary significantly, as it allows the model to leverage the actual dynamics without diluting them with padded states.
>
> **Memory Consumption**
>
> Regarding memory consumption, the total memory usage depends on the following:
>
> $• $ Number of Trajectories in the Buffer (Buffer Size): We use a fixed-size replay buffer, which holds $N$ trajectories.
>
> $• $ Average Length of Trajectories ($T$): Since trajectories have varying lengths, the memory usage will depend on the average trajectory length.
>
> $• $ Dimensionality of State and Action Spaces: Let $d_s$ be the dimension of the state, and $d_a$ be the dimension of the action. Each step in a trajectory stores both state and action information.
>
> $• $ Data type size: Denote by $n$ the specific data type size.
>
>
> The memory consumption ($M$) can be roughly estimated as:
> $
> M = NTn(d_s + d_a)
> $
>
> For example, if $N = 1000$ trajectories are stored, each with an average length of 200 steps, and assuming $d_s = 17$, $d_a = 6$, and using 32-bit floats (4 bytes), the memory requirement would be:
> $
> M = 1000 \times 200 \times (17 + 6) \times 4 \text{ bytes} = 18,400,000 \text{ bytes} \approx 18.4 \text{ MB}
> $
>
> This calculation provides an estimate, but the actual memory usage may vary depending on the environment and the specific implementation of the replay buffer.
>
> We will add more details about this implementation in supplementary material to clarify the structure of the replay buffer and the memory requirements.
>
>
> **Question 3:**
>
> Thank you for pointing out this observation regarding the variance of HalfCheetah increasing as training progresses.
>
> The increase in variance during longer training for the HalfCheetah environment can be attributed to several factors:
>
>
> $• $ Exploration-Exploitation Trade-off: In complex environments such as HalfCheetah, the agent continues to explore different strategies over time. As the policy becomes less constrained, the range of actions becomes more diverse, which can lead to higher variability in the returns, thereby increasing the variance.
>
> $• $ Environment Complexity and Instability: HalfCheetah is a highly dynamic and unstable environment. As the agent learns, it often employs increasingly aggressive and complex behaviors to maximize reward, which can lead to unstable trajectories and subsequently higher variance in performance.
>
> $• $ Replay Buffer with FIFO Strategy: The FIFO strategy in the replay buffer retains only the most recent trajectories. As training progresses, the diversity of these recent trajectories may lead to higher variations in sampled batches, which affects the stability of learning and contributes to increased variance.
>
> $• $ High Sensitivity of the Policy: As the policy approaches higher-performing regions of the solution space, it becomes more sensitive to the sampled experiences. Small differences in the sampled trajectories can lead to significant variations in policy updates, which can amplify the variance over time.
>
>
> We acknowledge that the increasing variance is a challenge, and potential approaches to mitigate this include prioritized replay strategies or adjustments to balance exploration and exploitation more effectively. We will add this discussion in the revised manuscript to clarify this behavior.

---

> ### Author Response · Authors · 2024-11-20
> **Response to Reviewer XcPF's comments (Continued)**
>
> **Question 4:**
>
> The definition and calculation of the explained variance will be addressed in our supplementary materials. To clarify further:
>
> The explained variance is calculated as follows:
>
>
> Explained Variance = $1 - \frac{\text{Var}(\hat{y} - y)}{\text{Var}(y)}$
>
>
> where $\hat{y}$ represents the predicted values from the critic network, and $y$ represents the ground truth returns. Specifically:
>
> $•$ Inputs ($\hat{y}$): These are the predicted state values generated by the value function approximation during training.
>
> $•$ Ground Truth ($y$): This is the cumulative return (discounted sum of rewards) obtained from the environment, calculated for each state within the trajectory.
>
> The explained variance metric provides insight into how well the value function explains the variance in the observed returns. A high explained variance implies that the value function is accurately capturing the variations in the true returns.
>
> We will update the supplementary materials to make the calculation and definition of explained variance more explicit and clear.

---

> ### Author Response · Authors · 2024-11-27
> **Response to Reviewer XcPF's comments (Continued)**
>
> We have now uploaded a newly revised version of our work. Please refer to the Summary for a detailed overview of all the changes we have made. If you have any additional questions or require further clarification, please do not hesitate to reach out. Thank you.

---

> ### Author Response · Authors · 2024-12-04
> **Response to Reviewer XcPF's comments**
>
> Dear Reviewer XcPF,
>
> We hope this message finds you well. Thank you again for your comments and feedback. We have carefully addressed all of your points in our revised version, and we would greatly appreciate it if you could take another look at the updated manuscript.
>
> While we understand that the discussion period has ended, we sincerely hope you will take our rebuttal and revised draft into consideration when finalizing your evaluation. We have done our best to address all the issues you raised and made significant improvements to resolve your concerns. Your feedback has helped us make the draft more technically solid and sound, and we are truly grateful for your contributions.
>
> Your insights have been extremely valuable to us, and we truly hope the improvements we have made meet your expectations. Thank you so much for your time and consideration.
>
> Best regards,
>
> The authors of Paper 11700

---

### Author Response · Authors · 2024-11-27
**Summary of the updated revision:**

We greatly appreciate all the insightful comments and feedback from the reviewers and would like to provide specific details on the revisions we have conducted based on the comments and suggestions from all reviewers, given the rebuttal period. Please see the following list of precise modifications in the revised draft.

$• $  Added Remark 5: We included a new remark to explain why we conducted sample efficiency analysis rather than exact sample complexity analysis, strengthening the theoretical foundation of our proposed algorithms. For more details, please refer to section A.4 in the Appendix. Additionally, we added a theoretical comparison between HP3O and HP3O+ in section A.5 to highlight how HP3O+ improves upon HP3O in terms of the policy improvement lower bound.

$• $  Expanded Comparative Evaluation: We added baseline methods (GEPPO and OffPolicy) and HP3O+ to the comparative evaluation in section 6.1. This addresses the major concern about the lack of sufficient baseline comparisons. Specifically, we implemented GEPPO, OffPolicy PPO (abbreviated as OffPolicy), and HP3O+ for all environments and incorporated the results in Figure 2. The discussion in section 6.1 has also been updated to reflect these additions. We included baseline methods (GEPPO and OffPolicy) and HP3O+ in the ablation studies in section 6.2. This revision is to extend the variance and time complexity plots to include more baselines and HP3O+, both Figure 3(a) and Figure 3(b). We also adjusted the discussion in section 6.2.

$• $  Enhanced Evaluation on Cartpole and Humanoid: We have added baselines methods (GEPPO and OffPolicy) and HP3O+ in Figure 5 for the Cartpole environment and the results for the Humanoid task in section A.6. We updated our evaluation on the Cartpole environment by adding GEPPO and OffPolicy results in Figure 5 in section A.6. Additionally, the result for the Humanoid task with different methods is also incorporated in this section. However, obtaining the result for the Humanoid task is quite time-consuming. Given the limited time within the rebuttal period, we could only obtain results for HP3O, SAC, A2C, and PPO, which still give us a preliminary evaluation of this complex continuous environment. For the rest of the methods, we will include them in the final version of this paper.

$• $ Converged Reward Table: Added converged reward table in section A.12. The converged performance of each algorithm is critical to evaluate them. Therefore, to address the comments from the reviewer and based on their suggestions, we have added a table (Table 2 in section A.12) to include the mean and standard deviation of the episodic reward at or close to the converged stage.

$• $ Episodic Return-Time Cost Analysis: Added figures in section A.13 to show the “episodic return-time cost of diverse methods. To show the precise wall-clock time of each method reaching a certain episodic return, we have added the results in Figure 12 in section A.13. However, due to the time limit, we only showed one environment: HalfCheetach, and will include results for all other environments in the final version.

$• $  Buffer Size and Memory Discussion: Added the technical discussion for the buffer size and memory calculation in section A.14. We added an extensive discussion on the impact of the buffer size on the model performance to reveal the relationship between replay buffer and model performance qualitatively. Additionally, we also provided the memory usage estimate in this section.

---

> ### Author Response · Authors · 2024-11-28
> **New Update**
>
> We have uploaded a new revised version that includes additional evaluation results. Section A.15 presents examples of evaluation outcomes for several environments.

---

### Meta-Review · Area_Chair_KwcR · 2024-12-22

**Metareview:**

This paper proposes a variation of PPO based on a trajectory buffer to reduce the noise in PPO and improve the learning efficiency. This paper is on a positive borderline. The main remaining concerns are the proposed scheme appears to be incremental and the theoretical and experimental results are limited. After reviewing the discussion, I think the novelty issue is minor. Nonetheless, I agree with reviewers (XcPF, zRpn, Ye97) more baselines are needed in the experiments, especially off-policy methods. The authors made some attempts to include results of other PPO variants, but other common baselines are needed to establish how the algorithm performs. In addition, another weakness of the paper is that only results on Mujoco problems are presented. I would encourage to also experiment with non-mujoco problems especially those with high-dimensional observations or discrete actions.

Considering these points, I would not recommend acceptance of the paper at its current status.

**Additional Comments On Reviewer Discussion:**

Reviewer XcPF raise concerns on incremental contribution, unclear justification, and missing (off-policy) baselines in experiments. The authors provide more details but I think the results and evaluation here are incremental as the reviewer mentioned. Reviewer zRpn raises issues of missing baselines (GePPO, Policy-on-off PPO  Off-policy PPO) in the experiments, missing important experimental results, and weak theoretical analysis. The response partially address the concerns; more baselines are still needed. Reviewer iV5K mentions issues of limited novelty, inconclusive experimental results and analyses; most concerns are addressed by the responses from the authors. Reviewer Ye97 raises concerns on missing (off-policy) baselines in the experiments, experiment design and potentially unsupported claims. While the rebuttal address some concerns, key experimental results with more baselines and more environments.

---

### Decision · Program_Chairs · 2025-01-22

Reject